# Tree Ensemble Explainability through the Hoeffding Functional Decomposition and TreeHFD Algorithm

**Clément Bénard**
Thales cortAIx-Labs - SINCLAIR AI Lab
1 avenue Augustin Fresnel, 91120 Palaiseau, France
clement-l.benard@thalesgroup.com

## Abstract

Tree ensembles have demonstrated state-of-the-art predictive performance across a wide range of problems involving tabular data. Nevertheless, the black-box nature of tree ensembles is a strong limitation, especially for applications with critical decisions at stake. The Hoeffding or ANOVA functional decomposition is a powerful explainability method, as it breaks down black-box models into a unique sum of lower-dimensional functions, provided that input variables are independent. In standard learning settings, input variables are often dependent, and the Hoeffding decomposition is generalized through hierarchical orthogonality constraints. Such generalization leads to unique and sparse decompositions with well-defined main effects and interactions. However, the practical estimation of this decomposition from a data sample is still an open problem. Therefore, we introduce the TreeHFD algorithm to estimate the Hoeffding decomposition of a tree ensemble from a data sample. We show the convergence of TreeHFD, along with the main properties of orthogonality, sparsity, and causal variable selection. The high performance of TreeHFD is demonstrated through experiments on both simulated and real data, using our `treehfd` Python package (https://github.com/ThalesGroup/treehfd). Besides, we empirically show that the widely used TreeSHAP method, based on Shapley values, is strongly connected to the Hoeffding decomposition.

## 1 Introduction

Tree ensembles have demonstrated remarkable predictive performance to tackle supervised learning problems with tabular data, over the past two decades. In particular, gradient boosted trees (Friedman, 2001; Chen and Guestrin, 2016) and random forests (Breiman, 2001) are probably the most successful algorithms to build tree ensembles. Recently, the extensive benchmark of Grinsztajn et al. (2022) has shown that tree ensembles still outperform deep neural networks on tabular data, and are therefore state-of-the-art on a wide range of problems. However, tree ensembles suffer from a major limitation with their lack of interpretability. Indeed, the prediction mechanism of a tree ensemble typically computes thousands of operations for a single prediction, making impossible to grasp how inputs are combined to generate predictions. Such black-box problem is shared by most machine learning algorithms, and is a serious obstacle when critical decisions are at stake, which is the case for industrial or healthcare applications, for example. The field of eXplainable AI, often called XAI, develops methods to explain predictions of learning algorithms, and has raised a strong interest in the community over the past few years (Guidotti et al., 2018; Speith, 2022; Amoukou, 2023). The two main approaches to obtain explainable algorithms are transparent models and post-hoc explanations. In the first case, the model structure is constrained to have a limited complexity, in order to keep a clear relation between the inputs and the output of the algorithm. However, transparent models usually have a limited accuracy, because of the strong constraints on their structure. The second type of methods provides explanations through post-treatments of the initial black-box model. A

popular example is variable importance, which quantifies the influence of each input variable on the algorithm output (Lundberg and Lee, 2017; Williamson et al., 2021). Although variable importance often provides highly relevant insights, it only gives partial information about the relation between the inputs and the output of the black box. On the other hand, functional decompositions have recently gained considerable momentum as an alternative approach to obtain more precise explainable methods than transparent models or variable importance (Bordt and von Luxburg, 2023; Hiabu et al., 2023; Idrissi et al., 2025). The principle is to break down a black-box model into a sum of functions with subsets of input variables as arguments. All functional components of one or two variables can be represented, and are therefore intrinsically transparent, while the accuracy of the initial black box is preserved. However, numerous functional decompositions can be obtained for the same black-box model, leading to different representations and interpretations. Therefore, the theoretical formulation of a functional decomposition must be clearly stated, to provide representations of the initial black box with clear properties, and enable precise conclusions about the underlying relation between the inputs and the output of the studied model. The Hoeffding decomposition (Hoeffding, 1948), extended by Stone (1994) and Hooker (2007), and Shapley values (Lundberg and Lee, 2017; Bordt and von Luxburg, 2023) provide theoretical frameworks to obtain such well-defined decompositions.

**Hoeffding functional decomposition.** The Hoeffding functional decomposition (HFD) breaks down a regression function into a sum of functions with variable subsets as arguments, and involves one functional component for each possible variable subset. Originally introduced in the seminal article of Hoeffding (1948) for independent input variables, the decomposition is unique and all functions are orthogonal. In the case of dependent input variables, a major breakthrough was done by Stone (1994) and Hooker (2007) to generalize the Hoeffding decomposition through hierarchical orthogonality constraints, which imply that two functions are orthogonal if one of the two variable subset arguments is included in the other one. Hence, the decomposition is still unique for dependent inputs, and a functional component is null if it is possible to break down the target function using only lower-order terms. This property provides a clear definition of interactions following the reluctance principle (Yu et al., 2019), and often leads to sparse decompositions essentially involving main effects and second-order interactions, which are intrinsically transparent. Later, Chastaing et al. (2012) and Idrissi et al. (2025) extended the validity of the decomposition for unbounded supports of the input distribution. Unfortunately, the practical estimation of the Hoeffding decomposition is a notoriously difficult problem (Hooker, 2007; Chastaing et al., 2012), and consequently, the HFD has long remained an abstract theoretical tool. Recently, Lengerich et al. (2020) proposed an algorithm to estimate this decomposition when the target function is a tree ensemble, and the input distribution is known. However, only a data sample is often available in practice, and the estimation of the input distribution for moderate or large dimensions is a very difficult task.

**Shapley values.** Shapley values build on game theory to define variable importance algorithms with attractive properties. Initially introduced by Owen (2014) and Lundberg and Lee (2017) for machine learning applications, Shapley values are now widely used to interpret both tree ensembles and neural networks. In particular, TreeSHAP (Lundberg et al., 2020; Yu et al., 2022; Muschalik et al., 2024b) is a fast algorithm to compute Shapley values for tree ensembles, and is implemented in the highly popular `xgboost` package. Recently, Herren and Hahn (2022), Bordt and von Luxburg (2023), and Hiabu et al. (2023) made strong connections between functional decompositions of black-box models and Shapley values—see the Supplementary Material. However, all these approaches inherit the estimation obstacles of Shapley values (Kumar et al., 2020), and heuristics with approximations are required to recover tractable algorithms, such as TreeSHAP with interactions, in the case of tree ensembles. Although TreeSHAP has become a highly popular and successful XAI method, it is also criticized for the lack of theoretical understanding of the estimated values (Amoukou, 2023, Chap. 3).

**Contributions.** The goal of this article is to introduce the TreeHFD algorithm to precisely estimate the Hoeffding decomposition of a tree ensemble, when only a data sample is available and the input distribution is unknown. Importantly, the theoretical analysis of TreeHFD shows the algorithm convergence, and exhibits the main properties of the obtained decomposition, providing a clear understanding of the resulting representation. To our best knowledge, TreeHFD is the first algorithm to provide accurate estimates of the Hoeffding functional decomposition in standard machine learning settings, where input variables are dependent and the input distribution is unknown. Additionally, we show that the functional decomposition induced by TreeSHAP is closely related to the HFD, through several experiments on both simulated and real data. This new result mitigates the main flaw of

TreeSHAP, since we connect the estimated values to well-defined theoretical quantities. However, TreeSHAP generates quite noisy functional components, and main effects can be entangled with interactions, which undermines the provided explanations. In the second section, the Hoeffding decomposition is formalized in the case of tree ensembles through a discretization of the orthogonality constraints, which enforces that the decomposition components are also unique and piecewise constant. These two characteristics are the cornerstone to build efficient estimates of the decomposition, without the input distribution. Then, we state the main properties of the HFD for tree ensembles, mainly about the component orthogonality, the sparsity of the decomposition, the causal variable selection, and its accuracy to approximate the HFD of the underlying regression function. In the third section, we introduce the TreeHFD algorithm, defined as a least square problem with a linear complexity with respect to the sample size, and implemented in the `treehfd` Python package. Then, we show the convergence of TreeHFD. Finally, we conduct extensive experiments in the last section, to show the high performance of TreeHFD, and the close connections between TreeSHAP and the HFD.

## 2 Hoeffding functional decomposition of tree ensembles

We first formalize the original Hoeffding decomposition of a square-integrable function in Theorem 1. While this decomposition has many valuable theoretical properties, its practical estimation from a data sample is still an open question, as explained in the introduction. However, Lengerich et al. (2020) take advantage of the piecewise constant form of tree ensembles to introduce an estimate of the HFD, provided that the distribution of the input variables is known. We overcome this limitation by introducing the Hoeffding decomposition of tree ensembles. We first formalize this decomposition in Theorem 2, and then derive its main theoretical properties.

### 2.1 Mathematical definition

We consider a standard supervised learning setting with a real random input vector $\mathbf{X}$ of dimension $p \in \mathbb{N}^\star$, and a real random output $Y$, defined in the following assumptions. We also denote by $\mathcal{V}_p = \{1, \ldots, p\}$ the variable index, by $\mathbf{X}^{(J)}$ the subvector of $\mathbf{X}$ with only the components in $J \subset \mathcal{V}_p$, and by $\mathcal{P}_p$ the set of all subsets of $\mathcal{V}_p$. Initially, Stone (1994) and Hooker (2007) defined the HFD for distributions of input variables with a hyperrectangle as support. In fact, it coincides with the theoretical analyses of tree ensembles, which also often assume this type of support for input distributions (Scornet et al., 2015; Wager and Athey, 2018), and is a mild assumption in practice. We therefore focus on such hyperrectangle supports, and take $[0,1]^p$ without loss of generality.

**Assumption 1.** *The input vector $\mathbf{X}$ takes values on $[0,1]^p$, and admits a density $f$ that is bounded by strictly positive constants $c_1, c_2 > 0$, that is, for all $\mathbf{x} \in [0,1]^p$, $c_1 \leq f(\mathbf{x}) \leq c_2$.*

**Assumption 2.** *The output $Y$ is defined by $Y = m(\mathbf{X}) + \varepsilon$, where $m : [0,1]^p \to \mathbb{R}$ is a square-integrable function, and $\varepsilon$ is an independent noise.*

The underlying regression function $m$ is estimated using a tree ensemble. More precisely, we denote by $T$ the prediction function of a tree ensemble built with $M \in \mathbb{N}^\star$ trees, from a given realization of a training dataset of fixed sample size, and from a realization of the tree randomizations. In the sequel, the function $T$ is thus considered as a deterministic function of a new query point $\mathbf{X}$, which avoids the repetitions that theoretical results are stated conditional on the training dataset. Furthermore, we define $T(\mathbf{X}) = \sum_\ell T_\ell(\mathbf{X})$, where $T_\ell$ is the prediction function of the $\ell$-th tree, multiplied by an aggregation coefficient. Such coefficient is typically the learning rate for boosted tree ensembles, or $1/M$ for random forests. We also denote by $\mathcal{T}_M = \{1, \ldots, M\}$ the tree index. Now, we state the original HFD introduced by Stone (1994) and Hooker (2007) in the context of Assumption 1.

**Theorem 1** (Hoeffding Decomposition (Stone, 1994; Hooker, 2007)). *If Assumption 1 is satisfied, and $\nu$ is a square-integrable real function defined on $[0,1]^p$, then there exists a unique set of functions $\{\nu^{(J)}\}_{J \in \mathcal{P}_p}$, such that for all $J \in \mathcal{P}_p$, $I \subset J$ with $I \neq J$, $\mathbb{E}[\nu^{(J)}(\mathbf{X}^{(J)})|\mathbf{X}^{(I)}] = 0$, and*

$$\nu(\mathbf{X}) = \sum_{J \in \mathcal{P}_p} \nu^{(J)}(\mathbf{X}^{(J)}).$$

In particular, if the input variables are further assumed to be mutually independent, the components of the decomposition take an explicit form as the Möbius transform of the conditional expectations, that is $\nu^{(J)}(\mathbf{X}^{(J)}) = \sum_{I \subset J}(-1)^{|J|-|I|}\mathbb{E}[\nu(\mathbf{X})|\mathbf{X}^{(I)}]$, for all $J \in \mathcal{P}_p$. However, this formula does

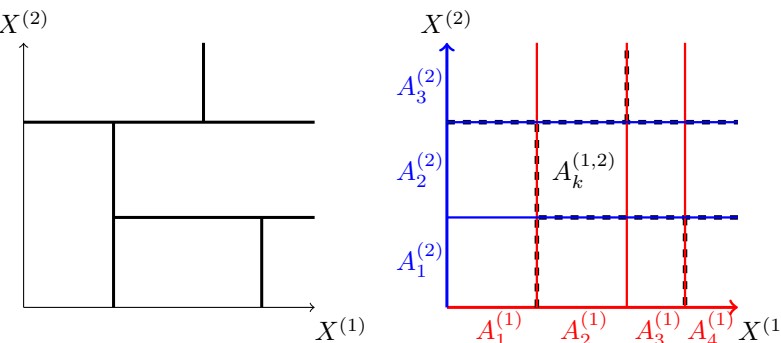

Figure 1: Example of the partition of $[0, 1]^2$ by a tree $T_\ell$ (left side), and the associated Cartesian tree partitions $\mathcal{A}_\ell^{(1)} = \{A_1^{(1)}, A_2^{(1)}, A_3^{(1)}, A_4^{(1)}\}$, $\mathcal{A}_\ell^{(2)} = \{A_1^{(2)}, A_2^{(2)}, A_3^{(2)}\}$, and $\mathcal{A}_\ell^{(1,2)}$ (right side).

not hold in the general case of dependent inputs, which makes the estimation of the Hoeffding decomposition especially difficult. In this general setting, Theorem 1 provides the HFD of the regression function $m(\mathbf{X})$ and the tree ensemble $T(\mathbf{X})$, respectively denoted by $\{m^{(J)}\}_{J \in \mathcal{P}_p}$ and $\{T^{(J)}\}_{J \in \mathcal{P}_p}$, which also depend on the distribution $f$ of the input variables $\mathbf{X}$. However, both decompositions are unknown in practice, since $m$ and $f$ are unknown, while only a data sample is available, and estimating $f$ is a very difficult problem. Our main goal is to provide an efficient explainability method of the tree ensemble $T$ through the HFD of the function $T$ for dependent input variables. Additionally, $T$ often estimates $m$ with a high accuracy, and therefore, the HFD of T also approximates the HFD of the function $m$, which is highly valuable to explain the relation between $\mathbf{X}$ and $Y$. Hence, we build an adaptation of the HFD tailored for tree ensembles. By construction, each tree of an ensemble is a piecewise constant function, and therefore discard the variability of the underlying regression function within each cell of the tree partition. Obviously, such approximation is necessary to obtain a good accuracy, since the trees are learned from a finite sample, which cannot capture the data patterns with an arbitrary high accuracy. In the same spirit, we apply a piecewise constant approximation to the distribution $f$ of the inputs $\mathbf{X}$. However, we use a more fine-grained partition than the original tree partition to efficiently handle the orthogonality constraints: the Cartesian tree partition, introduced in Definition 1 and illustrated in Figure 1.

**Definition 1** (Cartesian Tree Partition)**.** *For a tree $\ell \in \mathcal{T}_M$ and a variable $X^{(j)}$ with $j \in \mathcal{V}_p$, the values of the node splits involving $X^{(j)}$ are collected over all tree nodes to build a sequence of intervals $\mathcal{A}_\ell^{(j)}$, which forms a partition of $[0, 1]$. Then, for any $J \in \mathcal{P}_p$, $\mathcal{A}_\ell^{(J)} = \bigotimes_{j \in J} \mathcal{A}_\ell^{(j)}$ is the partition of $[0, 1]^{|J|}$, defined as the Cartesian product of the one-dimensional partitions. Clearly, the function $T_\ell$ is piecewise constant over $\mathcal{A}_\ell^{(\mathcal{V}_p)}$ (to lighten notations, $\mathcal{A}_\ell^{(\mathcal{V}_p)}$ is written $\mathcal{A}_\ell$ in the sequel).*

For a tree $\ell \in \mathcal{T}_M$, we define the tree HFD as the original HFD applied to the tree prediction function $T_\ell$, but with the density $f$ averaged over each cell of the Cartesian tree partition, instead of $f$. Theorem 2 below shows four critical properties of the tree HFD. First, the decomposition can still be written with the input vector $\mathbf{X}$ of distribution $f$, with a discretized version of the orthogonality constraints. More importantly, such a decomposition is unique for a given pair of tree $T_\ell$ and density $f$, the components of the decomposition are also piecewise constant over the Cartesian tree partitions, and the decomposition is independently defined for each tree. These three ingredients enable the design of consistent estimates of the tree HFD from a data sample, as we will see in the next section. Notice that all proofs are gathered in the Supplementary Material (S-Mat).

**Theorem 2** (HFD for Tree Ensembles)**.** *Let Assumption 1 be satisfied, $T = \sum_\ell T_\ell$ be the prediction function of a tree ensemble, and $\{\mathcal{A}_\ell^{(J)}\}_{J \in \mathcal{P}_p, \ell \in \mathcal{T}_M}$ the associated Cartesian tree partitions of Definition 1. Then, $T$ has a unique decomposition into a sum of functions $\{\eta_\ell^{(J)}\}_{J \in \mathcal{P}_p, \ell \in \mathcal{T}_M}$, such that $\eta_\ell^{(J)}$ is piecewise constant over $\mathcal{A}_\ell^{(J)}$ for all $J \in \mathcal{P}_p, \ell \in \mathcal{T}_M$, and*

$$T(\mathbf{X}) = \sum_{J \in \mathcal{P}_p} \eta^{(J)}(\mathbf{X}^{(J)}), \quad with \quad \eta^{(J)}(\mathbf{X}^{(J)}) = \sum_{\ell \in \mathcal{T}_M} \eta_\ell^{(J)}(\mathbf{X}^{(J)}) \ and \ T_\ell(\mathbf{X}) = \sum_{J \in \mathcal{P}_p} \eta_\ell^{(J)}(\mathbf{X}^{(J)}),$$

*and for all $\ell \in \mathcal{T}_M$, $J \in \mathcal{P}_p$, $I \subset J$ with $I \neq J$, $A \in \mathcal{A}_\ell^{(I)}$, $\mathbb{E}[\eta_\ell^{(J)}(\mathbf{X}^{(J)})|\mathbf{X}^{(I)} \in A] = 0$.*

Besides, Theorem 2 also holds when Assumption 1 is not satisfied. Indeed, since the number of terminal leaves is finite, the tree ensemble is constant in each direction outside an hyperrectangle,

and the tree HFD is also constant in this area. Therefore, even if the input distribution $f$ does not have a bounded support, Theorem 1 can be applied within this hyperrectangle, and the decomposition is simply extended outside with constant values. Furthermore, if the distribution $f$ takes null values, its average over each cell of a tree partition is strictly positive, as all terminal leaves contain as least one data point, and Theorem 1 can also be applied in this context to state the tree HFD.

## 2.2 Theoretical properties

**Orthogonality.** One of the main characteristics of the original HFD defined in Theorem 1 is that functions are hierarchically orthogonal, that is $\mathrm{Cov}[\nu^{(J)}(\mathbf{X}^{(J)}), \nu^{(I)}(\mathbf{X}^{(I)})] = 0$ for $I \subset J$ and $I \neq J$. In particular, this property clearly differentiates interaction terms from main effects in the decomposition. For example with $p = 2$, $\mathbb{E}[\nu(\mathbf{X})|X^{(1)}] = \nu^{(1)}(X^{(1)}) + \mathbb{E}[\nu^{(2)}(X^{(2)})|X^{(1)}]$, and $\nu^{(1,2)}$ vanishes by construction. Therefore, $\nu^{(1,2)}$ does not contain main effects, and is a pure interaction, as explained in Lengerich et al. (2020), and this property is obviously also true for higher-order interactions. In Theorem 3 below, we show that hierarchical orthogonality still holds for the tree HFD of a given tree of the ensemble, whereas it is not necessary the case for the global tree HFD components of the ensemble, because of the discretization across various tree partitions. However, we prove that orthogonality is almost satisfied, provided that the density $f$ of $\mathbf{X}$ does not vary too much within each cell of the tree partitions. We will show in the experimental section that orthogonality is almost satisfied in practice. To formalize the result, we denote by $\|\nu\|_\infty$ the usual infinite norm of a function $\nu$, and $\Delta_{\mathcal{A},f}$ the maximum variability of $f$ within each cell of the Cartesian tree partitions $\mathcal{A} = \{\mathcal{A}_\ell^{(J)}\}_{J \in \mathcal{P}_p, \ell \in \mathcal{T}_M}$. More precisely, we write $\Delta_{\mathcal{A},f} = \sup_{A \in \cup_\ell \mathcal{A}_\ell} \sup_{\mathbf{x},\mathbf{x}' \in A} |f(\mathbf{x}) - f(\mathbf{x}')|$.

**Theorem 3.** *Let Assumption 1 be satisfied, $T = \sum_\ell T_\ell$ be the prediction function of a tree ensemble, $\mathcal{A} = \{\mathcal{A}_\ell^{(J)}\}_{J \in \mathcal{P}_p, \ell \in \mathcal{T}_M}$ the associated Cartesian tree partitions, and $\{\eta_\ell^{(J)}\}_{J \in \mathcal{P}_p, \ell \in \mathcal{T}_M}$ the tree HFD defined in Theorem 2. We also consider $J \in \mathcal{P}_p$, and $I \subset J$ with $I \neq J$. Then, we have*

$$(i) \quad \forall \ell \in \mathcal{T}_M, \quad \mathrm{Cov}[\eta_\ell^{(J)}(\mathbf{X}^{(J)}), \eta_\ell^{(I)}(\mathbf{X}^{(I)})] = 0,$$

$$(ii) \quad \left| \mathrm{Cov}[\eta^{(J)}(\mathbf{X}^{(J)}), \eta^{(I)}(\mathbf{X}^{(I)})] \right| < \frac{c_2 - c_1}{c_1^2} \|\eta^{(I)}\|_\infty \left( \sum_{\ell=1}^M \|\eta_\ell^{(J)}\|_\infty \right) \Delta_{\mathcal{A},f}.$$

**Sparsity.** The HFD is sparse, since a functional component is included in the decomposition only if terms of lower orders are not sufficient to break down the target model. Therefore, we say that the HFD complies with the reluctance principle (Yu et al., 2019; Lengerich et al., 2020), and this is a consequence of hierarchical orthogonality. Theorem 4 below shows that the tree HFD preserves this important property, which often leads to sparse decompositions with essentially main effects and second-order interactions, which are intrinsically transparent.

**Theorem 4.** *Under the same assumptions as Theorem 3, if the functions $T_\ell$ can be written $T_\ell(\mathbf{X}) = g_\ell(\mathbf{X}^{(I)}) + h_\ell(\mathbf{X}^{(I')})$ for a pair of variable sets $I, I' \in \mathcal{P}_p$, with real functions $g_\ell, h_\ell$ for all $\ell \in \mathcal{T}_M$, then for all $J \in \mathcal{P}_p$ such that $J \not\subset I$ and $J \not\subset I'$, we have $\eta^{(J)}(\mathbf{X}^{(J)}) = 0$ a.s.*

**Causal variable selection.** We show that the tree HFD achieves the same causal variable selection than the decomposition induced by interventional Shapley values. Such decomposition is introduced by Bordt and von Luxburg (2023), through the generalization of Shapley values to all groups of variables. More precisely, Theorem 4 in Bordt and von Luxburg (2023) shows that each Shapley value function also defines a specific functional decomposition, called Shapley-GAM, which is in fact the Möbius transform of the value function (Osgood, 1957). The two most popular value functions are $v^{(obs)}(J, \mathbf{x}^{(J)}) = \mathbb{E}[T(\mathbf{X})|\mathbf{X}^{(J)} = \mathbf{x}^{(J)}]$ for $J \in \mathcal{P}_p$, defining observational SHAP (Lundberg and Lee, 2017), and $v^{(int)}(J, \mathbf{x}^{(J)}) = \mathbb{E}[T(\mathbf{x}^{(J)}, \mathbf{X}^{(-J)})]$, defining interventional SHAP (Datta et al., 2016; Janzing et al., 2020). While $v^{(obs)}(J, \mathbf{x}^{(J)})$ is the original proposal from Lundberg and Lee (2017), Janzing et al. (2020) show that interventional SHAP provides the causal effects of the input variables on the output $Y$, provided quite mild assumptions on the causal structure of the problem. However, Kumar et al. (2020) argue that both observational and interventional SHAP have strong drawbacks. For example, Theorem 4 does not hold for observational SHAP, since the dependence within input variables may result in non-null components for a variable $X^{(j)}$, even if T is constant with respect to $X^{(j)}$. On the other hand, interventional SHAP may be a poor approximation of the true data patterns, even if $T$ is highly accurate for a new dataset distributed as the training data. Indeed, because of the dependence between variables, some regions of the input space may have almost no data, and consequently, $T(\mathbf{X})$ may extrapolates with a low accuracy in these areas, on

which $v^{(int)}(J, \mathbf{x}^{(J)})$ highly relies through the marginal expectation (Hooker et al., 2021). In practice, a functional decomposition rarely involves all possible variable subsets, but rather a fraction of them. We define by $\Gamma$ the set of exogenous variables, which only have null components in a decomposition $\{\nu^{(J)}\}_{J \in \mathcal{P}_p}$, that is $\Gamma = \{j \in \mathcal{V}_p : \forall J \in \mathcal{P}_p, \nu^{(J \cup \{j\})} = 0\}$. In the following Theorem 5, we show that the set of exogenous variables is the same for tree HFD and interventional SHAP, which therefore achieves the same causal variable selection.

**Theorem 5.** *Under the same assumptions as Theorem 3, if $\Gamma_h$ is the set of exogenous variables for the tree HFD of $T_\ell$ with $\ell \in \mathcal{T}_M$, and $\Gamma_s$ for the decomposition of $T_\ell$ induced by interventional SHAP, then we have $\Gamma_h = \Gamma_s$.*

**HFD of the regression function.** Our main goal is to provide a functional decomposition of the tree ensemble $T$ with clear properties, as provided in the previous paragraphs. Furthermore, according to Assumption 2, the output $Y$ is defined by $Y = m(\mathbf{X}) + \varepsilon$, where $m$ is the underlying regression function. Therefore, $T$ is an estimate of $m$, and the tree HFD of $T$ also provides an approximation of the original HFD of the function $m$. In fact, Theorem 6 below shows that such approximation is of high accuracy if the Mean Square Error (MSE) of $T$ with respect to $m$ is small, and the distribution $f$ does not vary too much over each cell of the Cartesian tree partitions, that is $\Delta_{\mathcal{A}, f}$ is small. Tree ensembles often have a high accuracy in practice, as shown in the benchmark of Grinsztajn et al. (2022) for example, and the HFD of $T$ is thus often a good approximation of the HFD of $m$.

**Theorem 6.** *Let Assumptions 1 and 2 be satisfied, and $\{m^{(J)}\}_{J \in \mathcal{P}_p}$ be the HFD of $m(\boldsymbol{X})$. Then, there exists two constants $K_1, K_2 > 0$, such that for any tree ensemble $T = \sum_\ell T_\ell$ of Cartesian tree partitions $\mathcal{A}$ and tree HFD $\{\eta^{(J)}\}_{J \in \mathcal{P}_p}$, we have for $J \in \mathcal{P}_p$,*
$$\mathbb{E}[(\eta^{(J)}(\boldsymbol{X}^{(J)}) - m^{(J)}(\boldsymbol{X}^{(J)}))^2] \leq K_1 \mathbb{E}[(m(\boldsymbol{X}) - T(\boldsymbol{X}))^2] + K_2 \Delta_{\mathcal{A}, f}^2 \left( \sum_{\ell=1}^{M} \sqrt{\mathbb{E}[T_\ell(\boldsymbol{X})^2]} \right)^2.$$

In Corollary 1, we derive asymptotic conditions on the training of the tree ensemble $T$ to obtain a vanishing error of the tree HFD with respect to the HFD of $m$, inspired from (Meinshausen and Ridgeway, 2006). We recall that $T$ is a deterministic function, trained with a realization of the training dataset of fixed sample size, throughout the article. Here only, we denote by $n_T$ the size of this training dataset, and analyze the tree HFD accuracy when $n_T$ grows for a random training dataset.

**Corollary 1.** *Under the same assumptions as Theorem 6, if the input distribution $f$ is further assumed to be a Lipschitz function, the tree ensemble $T$ is $L_2$-consistent, all trees are bounded, the tree depth grows to infinity with the sample size $n_T$, the split of all tree nodes leaves at least a fraction $\gamma > 0$ of the observations in each child node, and the optimization of each node split is slightly randomized to have a positive probability to split with each variable, then we have for $J \in \mathcal{P}_p$, $\mathbb{E}[(\eta^{(J)}(\boldsymbol{X}^{(J)}) - m^{(J)}(\boldsymbol{X}^{(J)}))^2] \underset{n_T \to \infty}{\longrightarrow} 0$.*

## 3   TreeHFD algorithm

We first show that the tree HFD of Theorem 2 can be parametrized with a set of real coefficients, and is in fact the minimum of a convex loss function for each tree. Then, the TreeHFD algorithm is defined from an empirical version of this least square problem using a data sample. Finally, we prove the convergence of TreeHFD.

**A least square problem.** The tree HFD functions are piecewise constant on the Cartesian tree partitions. For each tree, such partition is easily constructed by collecting the split values of the tree nodes, and deriving the corresponding intervals. Then, for $\ell \in \mathcal{T}_M$, $J \in \mathcal{P}_p$, we estimate $\eta_\ell^{(J)}$ by a piecewise constant function $\mu_\ell^{(J)}$ of the form $\mu_\ell^{(J)}(\mathbf{x}^{(J)}) = \sum_{k=1}^{K_J} \beta_k^{(J)} \mathbb{1}_{\mathbf{x}^{(J)} \in A_k^{(J)}}$, for $\mathbf{x}^{(J)} \in [0, 1]^{|J|}$, where $\beta_1^{(J)}, \dots, \beta_{K_J}^{(J)} \in \mathbb{R}$ and $\{A_k^{(J)}\}_k$ are the $K_J$ cells of the Cartesian partition $\mathcal{A}_\ell^{(J)}$. Next, we define a loss function to quantify the violation of the orthogonality constraints and the error of the decomposition with respect to each tree of the ensemble. We use Lemma 4.1 from Hooker (2007) to simplify the conditions of orthogonality, transformed into the following constraints: for all $j \in J$ and $A \in \mathcal{A}_\ell^{(J \setminus j)}$, $\mathbb{E}[\eta_\ell^{(J)}(\mathbf{X}^{(J)}) | \mathbf{X}^{(J \setminus j)} \in A] = 0$. Then, we define the theoretical loss function $L^\star$ by
$$L^\star(\{\mu_\ell^{(J)}\}) = \mathbb{E}\Big[\big(T_\ell(\mathbf{X}) - \sum_{J \in \mathcal{P}_p} \mu_\ell^{(J)}(\mathbf{X}^{(J)})\big)^2\Big] + \sum_{J \in \mathcal{P}_p} \sum_{j \in J} \sum_{A \in \mathcal{A}^{(J \setminus j)}} \mathbb{E}[\mu_\ell^{(J)}(\mathbf{X}^{(J)}) \mathbb{1}_{\{\mathbf{X}^{(J \setminus j)} \in A\}}]^2,$$
and the tree HFD is the unique minimum of $L^\star$, where $L^\star = 0$. Given the form of the functions $\mu_\ell^{(J)}$, minimizing such loss function with respect to $\{\beta_k^{(J)}\}_{k, J}$ clearly boils down to a least square problem.

**Algorithm description.** The loss function $L^\star$ defined above has a number of terms growing exponentially with the dimension $p$, and seems intractable in practice. In fact, only few variables are involved in the splits of a shallow tree of a boosted ensemble. Consequently, only the associated subsets of variables are considered in the loss $L_n$ for each tree, and the other components of the tree HFD are null by definition. Additionally, we only consider HFD components involving variable interactions of limited order, and denote by $d_I$ this hyperparameter. While interactions of order two often occur in practice, it is rarely the case for interactions of higher orders. Therefore, $d_I$ is typically set to two by default, but it is obviously possible to set greater values in TreeHFD. Then, we define $\mathcal{P}_p^{(\ell)} \subset \mathcal{P}_p$, the set of all possible variable sets collected along the paths of the $\ell$-th tree, and of maximum size $d_I$. Overall, only a small fraction of the variable subsets are involved in the loss function of each tree, which can therefore be optimized at a small computational cost. Next, we assume that we have access to a data sample $\mathscr{D}_n = \{(\mathbf{X}_i, Y_i)\}_{i=1,\dots,n}$, independent of the training dataset of $T$. We define $L_n$, the empirical counterpart of $L^\star$ estimated with $\mathscr{D}_n$ for each tree—see proof of Lemma 6, by

$$
L_n = \frac{1}{n} \sum_{i=1}^{n} \Big( T_\ell(\mathbf{X}_i) - \sum_{J \in \mathcal{P}_p^{(\ell)}} \sum_{k=1}^{K_J} \beta_k^{(J)} \mathbb{1}_{\mathbf{X}_i^{(J)} \in A_k^{(J)}} \Big)^2
$$
$$
+ \sum_{J \in \mathcal{P}_p^{(\ell)}} \sum_{j \in J} \sum_{k=1}^{K_{J \setminus j}} \Big[ \Big( \frac{1}{n} \sum_{i=1}^{n} \mathbb{1}_{\{\mathbf{X}_i^{(J \setminus j)} \in A_k^{(J \setminus j)}\}} \Big)^{-\frac{1}{2}} \sum_{k'=1}^{K_J} \beta_{k'}^{(J)} \frac{1}{n} \sum_{i=1}^{n} \mathbb{1}_{\{\mathbf{X}_i^{(J)} \in A_{k'}^{(J)} \bigcap \mathbf{X}_i^{(J \setminus j)} \in A_k^{(J \setminus j)}\}} \Big]^2.
$$

Notice that for $J \in \mathcal{P}_p^{(\ell)}$ with $|J| = 1$, we get $J \setminus j = \emptyset$. In these cases, we set by convention that the involved indicator functions always take value one, and the corresponding terms in $L_n$ simply enforce that all HFD components have zero mean. Hence, the set of functions $\{\mu_{n,\ell}^{(J)}\}_J$ is defined as the minimum of $L_n$ through the optimization of the parameters $\{\beta_k^{(J)}\}_{k,J}$. Although the loss $L_n$ has a quite complex form, its computation only requires to collect the list of cells in which each data point falls. Then, simple counts give all the necessary coefficients to express $L_n$ as a quadratic function of the parameters $\{\beta_k^{(J)}\}_{k,J}$. In practice, we can use the available efficient software to solve quadratic programs. Indeed, the parameters $\{\beta_k^{(J)}\}_{k,J}$ are stacked together to form a single vector $\boldsymbol{\beta}$. Then, the loss $L_n$ is translated into a constraint matrix $\mathbf{C}_n$, where each square term gives a row of $\mathbf{C}_n$, and each element of the row is the coefficient of the corresponding $\beta_k^{(J)}$ in the square term of $L_n$. The target vector $\mathbf{Z}_n$ is defined by $T_\ell(\mathbf{X}_i)$ for the constraints of the first part of $L_n$, and by the null value for the orthogonality constraints. Also notice that many constraint rows are identical for the first part of $L_n$, since many data points fall in the same collection of cells, where $T_\ell$ is constant. These rows can thus be deduplicated and multiplied with appropriate weights to compact $\mathbf{C}_n$. Finally, the functions $\{\mu_{n,\ell}^{(J)}\}_J$ are defined from the parameters $\boldsymbol{\beta}_n^\star$, which solve $\boldsymbol{\beta}_n^\star = \mathrm{argmin}_{\boldsymbol{\beta}} \|\mathbf{Z}_n - \mathbf{C}_n \boldsymbol{\beta}\|_2^2$, where $\|\cdot\|_2$ is the Euclidean distance. Then, the least square problem is solved for each tree of the ensemble, and the estimated tree HFD component $\mu_n^{(J)}$ of $T$ is given by $\mu_n^{(J)} = \sum_{\ell=1}^{M} \mu_{n,\ell}^{(J)}$.

**Computational complexity.** The computational complexity of solving the above least square problem depends on the number of cells and variable subsets, and thus on the tree depth, but not on the sample size $n$ or directly on the dimension $p$. Additionally, the construction of the matrix $\mathbf{C}_n$ has a linear complexity with respect to the sample size $n$, and $\mathbf{C}_n$ is highly sparse by design, which greatly reduces the cost of the least square problem. Overall, the computational complexity of TreeHFD is linear with respect to $n$, does not depend on $p$ for shallow trees, and grows exponentially with the tree depth for large $p$. Therefore, TreeHFD is highly efficient for ensembles of shallow trees, such as boosted tree ensembles.

**Additional algorithm details.** We specify four additional details and extensions for TreeHFD algorithm. First, we can increase the efficiency of TreeHFD for deep trees that are typically involved in random forests, by introducing two hyperparameters $d_T$ and $d_V$ to control tree depth and the number of variable subsets. Since deep splits in random forest trees are often not significant (Duroux and Scornet, 2018), we use a preprocessing step before computing the Cartesian tree partitions, where node splits are pruned whenever they exceed a given tree depth threshold, provided by the hyperparameter $d_T$. Then, the original tree predictions are averaged in each cell of the new partition to define the new prediction function of the tree. Additionally, when the tree depth is large, the number of variable subsets $\mathcal{P}_p^{(\ell)}$ may become very high. Consequently, only the variable subsets at the first $d_V$ levels of the tree are extracted to build $\mathcal{P}_p^{(\ell)}$, to control its size, and thus the computational complexity of TreeHFD for deep trees, while focusing on the most influential variables. Secondly, by construction of the Cartesian tree partitions, some cells may contain no training data, since the

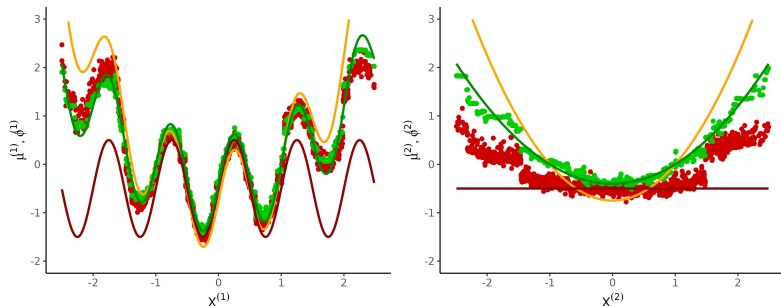

Figure 2: Main effects of the decompositions for $X^{(1)}$ and $X^{(2)}$. Solid lines provide the theoretical functions, with the HFD in green, int. SHAP in red, and obs. SHAP in orange. Green and red points are respectively the values provided by TreeHFD and TreeSHAP with interactions for `xgboost`.

Cartesian partitions are finer than the original tree ones. Hence, $L_n$ does not depend on the $\beta_k^{(J)}$ parameters of such empty cells. Once the reduced $\boldsymbol{\beta}_n^{\star}$ is computed, the parameter of each empty cell is set to the same value as the largest neighboring nonempty cell. For this specific reason, it is critical to break down the decomposition by tree, whereas merging all tree partitions with a unique cost function is highly inefficient. Thirdly, TreeHFD can also easily handle categorical and discrete variables, as fully explained in the S-Mat. Finally, TreeHFD algorithm also applies to classification problems for boosted tree ensembles, where the logit of each class is a continuous output that can be handled as the regression case. TreeHFD also applies to binary classification for all tree ensembles.

**Algorithm convergence.** We show the convergence of the TreeHFD algorithm towards the theoretical tree HFD formalized in Theorem 2, when the sample size grows. Therefore, TreeHFD provides a clear decomposition of the initial tree ensemble, which satisfies all the properties of orthogonality, sparsity, causal variable selection, and accuracy, stated in the theorems of the previous section. In the sequel, we show that these properties are satisfied in practice, through several batches of experiments. We highlight that Theorem 7 is stated for a deterministic tree ensemble $T$, trained with a fixed set of points, and for a growing size of $\mathscr{D}_n$, used to fit TreeHFD from $T$.

**Theorem 7.** *If Assumption 1 is satisfied, the hyperparameters $d_I$, $d_T$, and $d_V$ are set to the tree depth, $\{\mu_{n,\ell}^{(J)}\}_{J \in \mathcal{P}_p}$ is the minimum of $L_n$ over a compact set for $\ell \in \mathcal{T}_M$, and $\{\eta_\ell^{(J)}\}_{J \in \mathcal{P}_p}$ is the tree HFD of $T_\ell(\boldsymbol{X})$ defined in Theorem 2, then for all $J \in \mathcal{P}_p$, we have $\mu_{n,\ell}^{(J)} \xrightarrow{p} \eta_\ell^{(J)}$.*

## 4 Experiments

We first analyze an analytical case to better understand the behavior of TreeHFD, and then evaluate TreeHFD performance with real datasets, using `treehfd` package. The main competitor of TreeHFD is the decomposition induced by Shapley values, which is implemented in `xgboost` as an extension of TreeSHAP (Lundberg et al., 2020; Bordt and von Luxburg, 2023). Although the proposal of Lengerich et al. (2020) is a major step to estimate the HFD of tree ensembles, it requires to know the input distribution, and is therefore not adapted to our setting where only a data sample is available.

**Analytical case.** We consider a Gaussian random vector $\mathbf{X}$ of dimension $p = 6$, where each component has unit variance and all pairs of variables have the same correlation $\rho = 1/2$. Then, the output $Y$ is defined by $Y = m(\mathbf{X}) + \varepsilon$, where $m(\mathbf{X}) = \sin(2\pi X^{(1)}) + X^{(1)} X^{(2)} + X^{(3)} X^{(4)}$, and $\varepsilon \sim \mathcal{N}(0, 0.5^2)$. We study this analytical case because sinusoidal and linear functions are quite difficult to approximate with piecewise constant estimates, and it is possible to compute the analytical decompositions and introduce strong correlations within input variables, in the case of a Gaussian input vector. In the S-Mat, we state the analytical functional decompositions of $m(\mathbf{X})$ provided by the HFD, interventional SHAP, and observational SHAP. Notice that for the HFD, the correlation between $X^{(1)}$ and $X^{(2)}$ (respectively $X^{(3)}$ and $X^{(4)}$) introduces main effects for these variables, to purify interactions. As expected from Theorem 5, the same functional components are null for the HFD and interventional SHAP, whereas observational SHAP have non-null components for all variable subsets, because of the input dependence. It is clear that all components of observational SHAP mix all terms

Table 1: MSE for TreeHFD and TreeSHAP with respect to the specified target decompositions.

| Algorithm | Target | $\eta^{(1)}$ | $\eta^{(2)}$ | $\eta^{(3)}$ | $\eta^{(4)}$ | $\eta^{(5)}$ | $\eta^{(6)}$ | $\eta^{(1,2)}$ | $\eta^{(3,4)}$ | Others |
|---|---|---|---|---|---|---|---|---|---|---|
| TreeHFD - `xgboost` | HFD | 0.02 | 0.01 | 0.02 | 0.02 | 0.002 | 0.002 | 0.04 | 0.04 | $\leq 0.01$ |
| TreeHFD - RF | HFD | 0.15 | 0.02 | 0.01 | 0.03 | 0.004 | 0.003 | 0.10 | 0.29 | $\leq 0.01$ |
| TreeSHAP - `xgboost` | HFD | 0.06 | 0.23 | 0.13 | 0.12 | 0.002 | 0.002 | 0.35 | 0.27 | $\leq 0.02$ |
| TreeSHAP - `xgboost` | int. SHAP | 0.49 | 0.16 | 0.24 | 0.25 | 0.002 | 0.002 | 0.77 | 0.82 | $\leq 0.01$ |
| TreeSHAP - `xgboost` | obs. SHAP | 0.35 | 0.73 | 0.51 | 0.51 | 0.50 | 0.50 | 0.68 | 0.60 | NA |

Table 2: Dataset characteristics and performance of TreeHFD and TreeSHAP.

| Dataset | $n$ | $p$ | Accuracy xgboost | Residual MSE TreeHFD | Orthogonality TreeHFD | Orthogonality TreeSHAP | Local Variability TreeHFD | Local Variability TreeSHAP |
|---|---|---|---|---|---|---|---|---|
| Abalone | 4176 | 9 | 50% | 1% | 0.05 | 0.85 | 0.009 | 0.09 |
| Airfoil | 1503 | 5 | 95% | 2% | 0.02 | 0.24 | $2.10^{-5}$ | 0.04 |
| Bike Sharing | 17389 | 8 | 87% | 3% | 0.004 | 0.13 | $3.10^{-4}$ | 0.1 |
| Housing | 20640 | 8 | 83% | 1% | 0.05 | 0.55 | $7.10^{-4}$ | 0.05 |
| Concrete | 1030 | 8 | 94% | 0.4% | 0.003 | 0.33 | 0.01 | 0.05 |
| Nutrition | 2278 | 7 | 20% | 3% | 0.03 | 0.43 | 0.01 | 0.06 |
| Parkinson | 5875 | 19 | 96% | 1% | 0.07 | 0.18 | 0.02 | 0.1 |
| Power Plant | 9568 | 4 | 97% | 0.1% | NA | 0.9 | 0.006 | 0.02 |
| Superconduc. | 21263 | 81 | 92% | 1% | NA | NA | 0.007 | 0.2 |

of the original function $m$, and it is therefore difficult to understand the influence of each input from such a decomposition. Next, we fit a boosted tree ensemble on a training dataset $\mathscr{D}_n$ of size $n = 5000$, using `xgboost` with $M = 100$ trees, and the default value for the other parameters. Then, we fit TreeHFD on the tree ensemble using $\mathscr{D}_n$, and display the result in Figure 2. Finally, we estimate the Mean Square Error (MSE) of each TreeHFD component, using the analytical formulas and an independent testing dataset. Besides, we perform the same experiment with TreeSHAP, and also using random forests (RF) instead of `xgboost`. Results are averaged over ten repetitions, and reported in Table 1, where standard deviations are negligible and thus omitted. Hence, it is clear from Table 1 that TreeHFD based on `xgboost` provides highly accurate estimates of the HFD, and outperforms the decomposition provided by TreeSHAP. This result is quite expected since TreeSHAP is not explicitly designed to target the HFD. However, Table 1 also highlights a surprising phenomenon: TreeSHAP more accurately estimates the HFD than the theoretical decompositions induced by interventional SHAP and observational SHAP. This implies that TreeSHAP has strong connections with the HFD, and the real data cases will further confirm this trend. Importantly, the obtained TreeHFD components are almost hierarchically orthogonal, with correlation coefficients between interactions and the associated main effects of about $0.05$ or smaller. We provide all experimental settings in the S-Mat, including additional experiments with various sample sizes, tree depths, and comparisons with `glex` (Hiabu et al., 2023) and EBM (Nori et al., 2019). In particular, we show that TreeHFD is not very sensitive to tree depth, and thus remains highly efficient for shallow trees.

**Real data cases.** We assess the performance of TreeHFD using nine real public datasets from the UCI repository (Kelly et al., 2024), with the main characteristics summarized in Table 2—see the S-Mat for all details about the displayed metrics. We first show that TreeHFD with interactions of second order (i.e. $d_I = 2$) successfully estimates the HFD of the fitted `xgboost` models, since the TreeHFD residual MSE is about $1\%$ of the output variance for all tested datasets, as reported in Table 2, and hierarchical orthogonality is well approximated by TreeHFD. Indeed, we compute the correlation coefficients between the interaction components and their associated main effects, and report the maximum absolute value for each dataset in Table 2, for both TreeHFD and TreeSHAP. Hence, the results show that the maximum absolute correlation for each dataset is about $0.05$ for TreeHFD or smaller, whereas it is frequently above $0.5$ for TreeSHAP. Therefore, TreeHFD components are almost hierarchical orthogonal, whereas main effects and interactions can be strongly entangled for TreeSHAP. By construction, TreeHFD inherits the eventual overfitting of the initial tree ensemble, as for the "Nutrition" dataset. Furthermore, we show that TreeSHAP generates quite noisy decompositions, which can be problematic for local interpretations. Indeed, a slight perturbation of

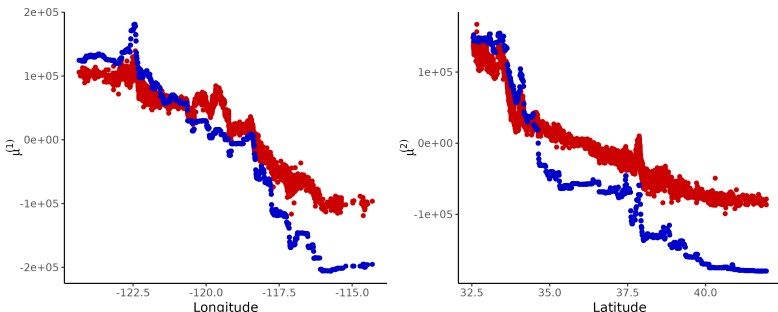

Figure 3: For the "Housing" dataset, main effects of "Longitude" and "Latitude" in the decompositions of respectively TreeHFD in blue and TreeSHAP with interactions in red.

the input may result in a large change in the output decomposition values. This phenomenon clearly undermines the local explanations provided by TreeSHAP. We quantify local variability as the mean variance of a functional component over the nearest neighbors of each point, and averaged over all main components—see the S-Mat for a formal definition. The obtained metric is reported in Table 2 and shows that TreeHFD is locally much more stable than TreeSHAP for all tested cases.

**Housing dataset.** We focus on the California Housing dataset (Kelly et al., 2024), which gathers housing prices with various characteristics. We fit `xgboost` and TreeHFD, and display the main resulting components in Figure 3. We see that the "Longitude" component of TreeHFD clearly identifies the peak in housing prices, corresponding to the San Francisco Bay Area (-122°25'), whereas TreeSHAP does not really detect it. Indeed, TreeSHAP decomposition is quite noisy, and the main effect of the "Longitude" variable is entangled with the interaction between "Longitude" and "Latitude" variables. For TreeSHAP, this interaction function has a variance of $16\%$ of the total variance $\mathbb{V}[T(\mathbf{X})]$ of the tree ensemble output, whereas it is only $4\%$ in the case of TreeHFD, because interactions are purified. From another perspective, the absolute correlation coefficients between the two main effects and the interaction are about $0.05$ for TreeHFD, close to the targeted null value of hierarchical orthogonality, whereas it is about $0.20$ for TreeSHAP. Additionally, TreeSHAP hides in this interaction that house prices are lower in northern and eastern California, as shown by TreeHFD.

**Additional datasets.** The same analysis for the "Bike Sharing" and "Superconductivity" datasets (Kelly et al., 2024) is provided in the S-Mat. In the first case, the dataset gathers the number of bikes rented by hour from a US bike renting system, along with time and weather information. TreeHFD and TreeSHAP generate close decompositions of the bike rentals, but SHAP values are quite noisy, and can therefore be misleading when predictions are analyzed one by one. The identified patterns can be easily interpreted. For example, there are renting peaks at the beginning and the end of the day, when people commute to work, and a strong interaction between the variables "Hour" and "Week day", since the peak of bike rentals is in the middle of the day during the weekend. There is also a drop of bike rentals when it is too hot or too cold, as expected. Finally, the "Superconductivity" dataset shows the high performance of TreeHFD for a quite large input dimension of $81$.

## 5   Conclusion

We have introduced the TreeHFD algorithm to estimate the Hoeffding decomposition of a tree ensemble from a data sample, which therefore provides an efficient explainability method. To our best knowledge, TreeHFD is the first algorithm to accurately estimate the HFD in standard machine learning settings, where only a data sample is available. Furthermore, we have empirically established strong connections between TreeSHAP and the HFD. In future work, conducting a theoretical analysis of this connection will deepen the understanding of the widely used TreeSHAP method.

**Limitations.** The main limitation of TreeHFD is the high computational complexity for large tree depths, which can be efficiently mitigated using pruning, controlled by two hyperparameters. Additionally, TreeHFD requires the access to the internal tree structures, as provided by most open source software, but not always by proprietary tools.

## Acknowledgments and Disclosure of Funding

We thank the reviewers for their insightful comments and suggestions. Clément Bénard has received support from the GATSBII project (ref. ANR-24-CE23-6645), funded by the ANR (french National Research Agency), and the FaRADAI project (ref. 101103386), funded by the European Commission under the European Defence Fund (EDF-2021-DIGIT-R).

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

# Supplementary Material (S-Mat)

## A  Categorical and discrete variables

Discrete and categorical variables are straightforward to handle in TreeHFD algorithm, even if we focus on continuous inputs for the sake of clarity. Indeed, discrete variables are considered as numeric ones by the initial tree ensemble. For example, if a variable $X^{(1)}$ takes values in $\{0.2, 0.4, 0.8\}$, a tree can typically build the partition $\{[0, 0.3], [0.3, 0.6], [0.6, 1]\}$, with splits at the middle of two values of $X^{(1)}$. Then, the associated distribution is constant in each cell, and the values are given by the weights of the initial discrete distribution. For categorical variables, we follow the approach of two of the main tree ensemble implementations, `xgboost` (Chen and Guestrin, 2016) for boosted trees and `ranger` (Wright and Ziegler, 2017) for random forests, using one-hot-encoding to get binary discrete variables, naturally handled by TreeHFD as explained before.

## B  Additional experiments and settings

Experiments were conducted with a standard computer machine with Ubuntu OS and the following main characteristics: Intel Core i5 CPU (2.30 GHz) with 6 cores and 16 GB of RAM. Notice that we use `xgboost` software in the experiments, in accordance with its Apache License 2.0.

### B.1  Competitors

**Shapley values.**   As mentioned in the introduction, Herren and Hahn (2022), Bordt and von Luxburg (2023), and Hiabu et al. (2023) made strong connections between functional decompositions of black-box models and Shapley values. In particular, Bordt and von Luxburg (2023) introduce $n$-Shapley values as a generalization to all subgroups of variables, which then induces a functional decomposition. On the other hand, Herren and Hahn (2022) and Hiabu et al. (2023) respectively show how to connect observational SHAP and interventional SHAP to functional decompositions with practical estimates. Additionally, Linear TreeSHAP (Yu et al., 2022) and TreeSHAP-IQ (Muschalik et al., 2024b,a) improve the computational complexity of the original TreeSHAP implementation (Lundberg et al., 2020), and target the same quantities. TreeSHAP-IQ can also provide high-order interactions.

Overall, these algorithms target observational SHAP or interventional SHAP, but not the Hoeffding functional decomposition, and are therefore not direct competitors of TreeHFD, with the notable exception of TreeSHAP, as discussed in the article. Although SHAP values have demonstrated great success to explain machine learning algorithms, Kumar et al. (2020) show that both observational SHAP and interventional SHAP have quite strong limitations, while the HFD is an interesting alternative to overcome these problems. In particular, TreeHFD shares the causal variable selection of interventional SHAP, but is based on the original distribution of the inputs, whereas interventional SHAP may extrapolate in areas where the tree ensemble has a low accuracy. Furthermore, observational SHAP introduces functional components for exogenous variables because of the input dependence, and can thus generate decompositions with a very high number of components, which are hardly interpretable.

In the next section, we run the analytical case with the algorithm from Hiabu et al. (2023), implemented in the R-package `glex` available online, which provides efficient estimates of the functional decompositions induced by interventional SHAP, as we will see. We do not run experiments with the promising TreeSHAP-IQ algorithm because it computes the same quantities as TreeSHAP implemented in `xgboost` for main effects and second-order interactions, and the computational time of TreeSHAP-IQ is high. Indeed, although its computational complexity is highly improved over the original TreeSHAP, TreeSHAP-IQ is designed to explain a specific prediction, and consequently, predictions are not vectorized in `shapiq` implementation (Muschalik et al., 2024a), which does not use a compiled language. Therefore, `shapiq` is efficient to compute a single prediction, which takes only a few seconds for the analytical case (with the machine described at the beginning of the section), but running all predictions one by one with $n = 5000$ takes several hours for one of the ten repetitions, while it takes less than one minute with our `treehfd` implementation or TreeSHAP with `xgboost`.

Table 3: Analytical functional decompositions ($C = \exp(-2\pi^2(1 - \rho^2))$). For $|J| = 1$, functions have $x$ as arguments, and $(x, z)$ for $|J| = 2$.

| $J$ | HFD | Interventional SHAP | Observational SHAP |
|---|---|---|---|
| $\emptyset$ | $2\,\rho$ | $2\rho$ | $2\rho$ |
| $\{1\}$ | $\sin(2\pi x) + \frac{\rho}{1+\rho^2}(x^2 - 1)$ | $\sin(2\pi x) - \rho$ | $\sin(2\pi x) + \rho(1+\rho)(x^2 - 1)$ |
| $\{2\}, \{3\}, \{4\}$ | $\frac{\rho}{1+\rho^2}(x^2 - 1)$ | $-\rho$ | $C\sin(2\pi\rho x) + \rho(1+\rho)(x^2 - 1)$ |
| $\{5\}, \{6\}$ | $0$ | $0$ | $C\sin(2\pi\rho x) + 2\rho^2(x^2 - 1)$ |
| $\{1,2\}, \{3,4\}$ | $\rho\frac{1-\rho^2}{1+\rho^2} - \frac{\rho}{1+\rho^2}(x^2 + z^2) + x \times z$ | $\rho + x \times z$ | $\phi^{(1,2)}, \phi^{(3,4)} \neq 0$ |
| Others | $0$ | $0$ | $\phi^{(J)} \neq 0$ |

**Generalized additive models.** The scope of the article is to estimate the HFD of black-box tree ensembles from a data sample, to provide an efficient post-hoc XAI method. Therefore, generalized additive models (GAMs) are out of the article scope, since they build transparent models in the first place. However, they also provide models in the same form of sums of low-order functions of the input variables. Hence, we mention two powerful and widely used GAMs: Explainable Boosting Machines (EBM) (Nori et al., 2019), and GAMI-NET (Yang et al., 2021). Both methods exhibit high predictive performance, quite often on par with boosted tree ensembles, but their targeted theoretical decomposition is unknown, which makes their interpretation more difficult.

EBM also takes advantage of boosted trees to provide GAMs with interactions of order two. In the next section, we run experiments to show that EBM can also estimate the HFD with a high accuracy. In a similar way as TreeSHAP, EBM seems to have strong connections with the HFD. However, we will see in the analytical case that EBM can introduce functional components for exogenous variables, because of the input dependence. Therefore, EBM does not share the sparsity and causal variable selection properties of TreeHFD and TreeSHAP. EBM can also generate different decompositions than TreeHFD, and also face instability issues when the input dimension is quite large, as shown in the experiments with the "Superconductivity" dataset in Subsection B.3.

GAMI-NET uses neural networks to also provide GAMs with second-order interactions. However, interactions are included only if variables have main effects. Although experiments show that GAMI-NET is highly accurate (Yang et al., 2021), it often does not target the HFD. For example, in the analytical case, if the Gaussian input vector is slightly modified to make $X^{(3)}$ and $X^{(4)}$ independent, then these two variables do not have main effects, and their interaction term will be set as null in GAMI-NET, instead of $X^{(3)}X^{(4)}$ in the updated HFD.

## B.2 Analytical case

**Analytical decompositions.** In Table 3, we provide the analytical functional decompositions of $m(\mathbf{X})$ provided by the HFD, interventional SHAP, and observational SHAP (Bordt and von Luxburg, 2023, Theorem 4). For the sake of clarity, some formulas are not displayed in the table and are given below. As mentioned in the article, the correlation between $X^{(1)}$ and $X^{(2)}$ introduces a main effect for these two variables in the HFD, which is of increasing magnitude with the correlation $\rho$, whereas the variability of $\eta^{(1,2)}$ reduces as $\rho$ increases. This is a consequence of the orthogonality constraints leading to pure interactions in the HFD, since $X^{(1)}$ or $X^{(2)}$ alone can partially estimate the term $X^{(1)}X^{(2)}$, because of their correlation. Symmetrically, variables $X^{(3)}$ and $X^{(4)}$ have the same behavior. Notice that constant main effects are also introduced for these variables in the case of interventional SHAP. As claimed in the article, the decomposition of observational SHAP is hard to interpret, since all components mix all terms of the initial function, because of the input dependence.

The main interactions of observational SHAP decomposition are given by

$$\phi^{(1,2)}(x, z) = C_0 + C_{main}(x^2 + z^2) + C_{inter}x \times z - C_\rho\sin(2\pi\rho x)$$

$$\phi^{(3,4)}(x, z) = C_0 + C_{main}(x^2 + z^2) + C_{inter}x \times z + C'_\rho\sin\left(\frac{2\pi\rho}{1+\rho}(x + z)\right)$$

$$- C_\rho\sin(2\pi\rho x) - C_\rho\sin(2\pi\rho z),$$

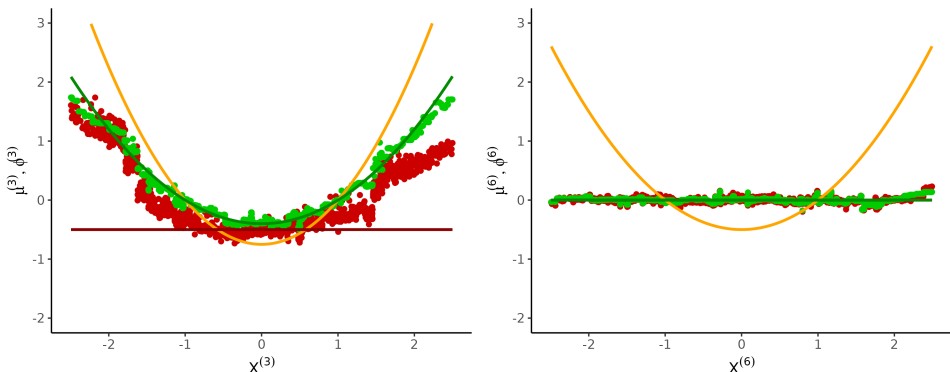

Figure 4: For the analytical case, main effects of the decompositions for $X^{(3)}$ and $X^{(6)}$. Solid lines provide the theoretical functions, with the HFD in green, int. SHAP in red, and obs. SHAP in orange. Green and red points are respectively the values provided by TreeHFD and TreeSHAP with interactions for `xgboost`.

Table 4: Cumulated MSE of the HFD components for TreeHFD and TreeSHAP with increasing sample sizes, for the analytical case (with the standard deviations in brackets).

| Sample size $n$ | 100 | 500 | 1000 | 2000 | 3000 | 5000 | 10000 |
|---|---|---|---|---|---|---|---|
| TreeHFD | 1.94 (0.4) | 0.79 (0.1) | 0.55 (0.05) | 0.37 (0.03) | 0.28 (0.02) | 0.21 (0.02) | 0.14 (0.007) |
| TreeSHAP | 2.38 (0.7) | 1.32 (0.2) | 1.26 (0.09) | 1.20 (0.09) | 1.20 (0.08) | 1.24 (0.10) | 1.27 (0.08) |

where

$$C_0 = 2\rho^2 + \frac{\rho}{1+2\rho}\Big(1 - \frac{2\rho^2}{1+\rho}\Big), \qquad C_{main} = \frac{\rho^2}{(1+\rho)^2} - \rho(1+\rho),$$

$$C_{inter} = 1 + \frac{2\rho^2}{(1+\rho)^2}, \qquad C_\rho = e^{-2\pi^2(1-\rho^2)}, \qquad C'_\rho = e^{-2\pi^2(1-\frac{2\rho^2}{1+\rho})}.$$

We do not provide the other high-order interaction terms for observational SHAP to avoid tedious formulas, since these functions are not really meaningful and take the same form as $\phi^{(1,2)}$ and $\phi^{(3,4)}$.

**Additional figures.** We provide additional figures for the main case of TreeHFD combined with `xgboost`, presented in the article in Section 4 for $n = 5000$, $M = 100$ trees, and the default values for the other `xgboost` parameters. Notice that the obtained tree ensemble has a proportion of output explained variance of $90\%$ (noise variance is $5\%$ of $\mathbb{V}[Y]$), and that the TreeHFD residual MSE is $2\%$ of $\mathbb{V}[T(\mathbf{X})]$. Hence, Figure 4 provides the main effects estimated by TreeHFD and TreeSHAP for $X^{(3)}$ and $X^{(6)}$, and Figure 5 gives the interaction function of $X^{(1)}$ and $X^{(2)}$ estimated by TreeHFD. We finally specify that one run of TreeHFD for this analytical case with $n = 5000$ and $M = 100$ trees takes less than 1 minute with our `treehfd` implementation.

**Sample size.** We repeat the same experiments for the analytical case of Section 4 with the sample size varying from $n = 100$ to $n = 10000$. We report the results in Table 4, which displays the cumulated MSE over all components for TreeHFD and TreeSHAP based on `xgboost`, for each sample size. In the case of TreeHFD, the cumulated MSE decreases as the sample size increases, as expected from the convergence result of Theorem 7, whereas TreeSHAP MSE is quite constant for $n \geq 1000$. This suggests that TreeSHAP does not converge towards the HFD, although it more accurately estimates the HFD than the decomposition induced by observational and interventional SHAP.

**Tree depth.** TreeHFD relies on the discretization provided by the Cartesian tree partitions. The number of cells for each tree is a direct consequence of the tree depth. Hence, we analyze the cumulated MSE for the analytical case with `xgboost` and the settings of Section 4, but varying tree depths (instead of the default value of 6 used in the article). The results are displayed in Table 5, and

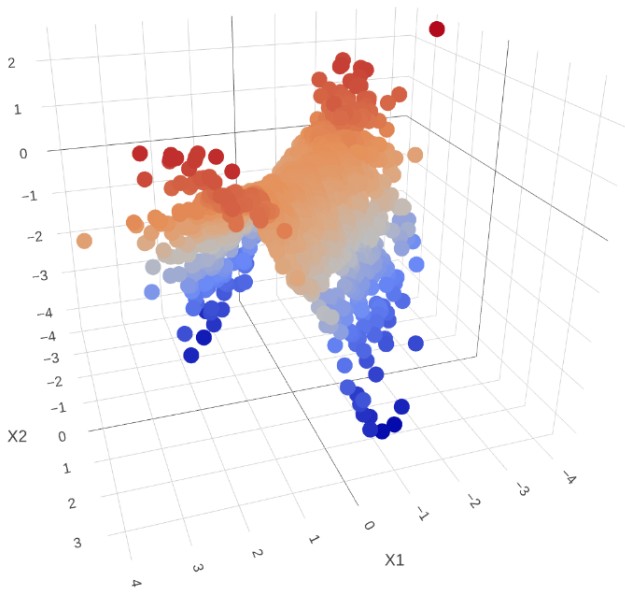

Figure 5: For the analytical case, interaction function of $X^{(1)}$ and $X^{(2)}$ estimated by TreeHFD.

Table 5: Cumulated MSE of the HFD components for TreeHFD with `xgboost` for various tree depths, for the analytical case (with the standard deviations in brackets).

| Tree depth | 1 | 2 | 3 | 4 | 5 | 6 | 7 |
|---|---|---|---|---|---|---|---|
| Cumulated MSE | 1.49 (0.05) | 0.34 (0.02) | 0.26 (0.2) | 0.20 (0.02) | 0.19 (0.01) | 0.20 (0.01) | 0.26 (0.02) |

show that even for very shallow trees (a depth of 3 for example), TreeHFD performance is high, and is overall not very sensitive to the tree depth. Only in the case of a tree depth of 1, interactions are not estimated and the cumulated MSE is strongly degraded.

**TreeHFD with random forests.** As briefly stated in the article, we perform the same experiment using random forests with the `ranger` implementation instead of `xgboost`, combined with TreeHFD, and results are reported in Table 1. The number of trees is still set to 100, and tree depth is limited to ten, i.e. $d_T = d_V = 10$ in TreeHFD, since the reduction of the forest accuracy is small compared to fully grown trees for this value. We observe that the accuracy is reduced in the case of TreeHFD based on random forests compared to `xgboost` for the main effect of $X^{(1)}$ involving a sinusoidal function and also for the interaction components. Such behavior is not very surprising because of the greedy forest construction, which considers variable one by one for the split optimizations.

**Comparisons with `glex`.** We run the analytical case with the same settings than in the article using the algorithm from Hiabu et al. (2023), implemented in the R-package `glex`. The results are displayed in Table 6, along with the values of Table 1 obtained for TreeHFD and TreeSHAP to enable comparisons. We do not display standard deviations, since they are negligible compared to the raw MSE values, as in Table 1. As expected, `glex` is an efficient estimate of the decomposition induced by interventional SHAP, and strongly outperforms TreeSHAP for this specific task. Consequently, `glex` does not provide good estimates of the HFD. The MSE of `glex` for interventional SHAP is similar to the MSE of TreeHFD for the HFD for main effects. However, the MSE of `glex` is higher for interaction components than the MSE of TreeHFD. As already mentioned, interventional SHAP is more difficult to estimate accurately than the HFD, because the value function involves expectations with respect to input distributions that are different from the original one (Kumar et al., 2020).

**Comparisons with EBM.** We run the analytical case with the same settings than in the article, using the EBM algorithm (Nori et al., 2019) with the default parameters. The results are displayed in Table 6, and show that EBM has an accuracy close to TreeHFD to estimate the HFD, even slightly

Table 6: MSE for `glex`, EBM, TreeHFD, and TreeSHAP with respect to the specified target decompositions, for the analytical case.

| Algorithm | Target | $\eta^{(1)}$ | $\eta^{(2)}$ | $\eta^{(3)}$ | $\eta^{(4)}$ | $\eta^{(5)}$ | $\eta^{(6)}$ | $\eta^{(1,2)}$ | $\eta^{(3,4)}$ | Others |
|-----------|--------|--------------|--------------|--------------|--------------|--------------|--------------|----------------|----------------|--------|
| TreeHFD - `xgboost` | HFD | 0.02 | 0.01 | 0.02 | 0.02 | 0.002 | 0.002 | 0.04 | 0.04 | $\leq 0.01$ |
| EBM | HFD | 0.03 | 0.01 | 0.01 | 0.01 | 0.02 | 0.01 | 0.03 | 0.03 | $\leq 0.01$ |
| `glex` - `xgboost` | HFD | 0.49 | 0.47 | 0.46 | 0.47 | 0.003 | 0.003 | 1.2 | 1.2 | $\leq 0.02$ |
| TreeSHAP - `xgboost` | int. SHAP | 0.49 | 0.16 | 0.24 | 0.25 | 0.002 | 0.002 | 0.77 | 0.82 | $\leq 0.01$ |
| `glex` - `xgboost` | int. SHAP | 0.03 | 0.01 | 0.01 | 0.01 | 0.003 | 0.003 | 0.13 | 0.12 | $\leq 0.02$ |

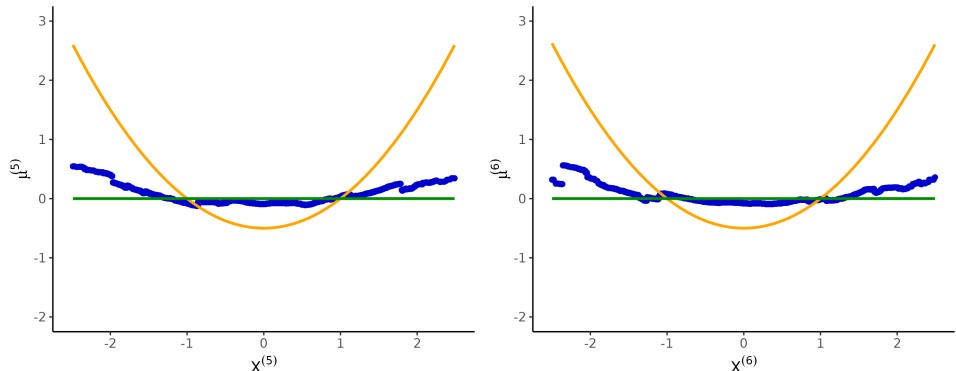

Figure 6: For the analytical case, main effects of EBM decompositions for $X^{(5)}$ and $X^{(6)}$. Solid lines provide the theoretical functions, with the HFD in green and obs. SHAP in orange (int. SHAP is also null and hidden by the HFD). Blue points are the values provided by EBM decomposition.

better for several components, but uses 25000 boosting rounds instead of the arbitrary number of 100 rounds for the `xgboost` model, broken down by TreeHFD. More importantly, the two main effect components of $X^{(5)}$ and $X^{(6)}$ that are null in the analytical HFD, have a MSE ten times higher for EBM than for TreeHFD and TreeSHAP. The components generated by EBM are displayed in Figure 6, which shows that EBM introduces non-null functions for these components, because of the correlation between $X^{(5)}$ and $X^{(6)}$ and the other input variables. We deepen this analysis by running the analytical case with EBM for increasing sample sizes. Table 7 provides the cumulated MSE for $X^{(5)}$ and $X^{(6)}$ components for both EBM and TreeHFD for these various sample sizes, averaged over ten repetitions. While the MSE of TreeHFD decreases as the sample size increases as expected from Theorems 4, 5, and 7, the MSE of EBM is quite constant. Therefore, it seems that EBM does not converge towards the HFD, and does not share the properties of sparsity and causal variable selection that TreeHFD and TreeSHAP exhibit.

## B.3  Real data cases

We first provide the definitions of the metrics involved in Table 2. Then, we analyze the "Superconductivity" and "Bike sharing" datasets. The first one shows the high performance of TreeHFD with a quite large input dimension, while the second dataset is an example where TreeHFD and TreeSHAP generate close decompositions. Therefore, the decompositions of the two algorithms can be similar or different depending on the data properties.

Table 7: Cumulated MSE of $X^{(5)}$ and $X^{(6)}$ components for EBM and TreeHFD with respect to the HFD, for the analytical case with an increasing sample size (standard deviations in brackets).

| Sample size $n$ | 10000 | 20000 | 30000 | 40000 |
|-----------------|-------|-------|-------|-------|
| TreeHFD - `xgboost` | $0.0020$ $(4.10^{-4})$ | $0.0011$ $(1.10^{-4})$ | $6.1 \times 10^{-4}$ $(1.10^{-4})$ | $5.5 \times 10^{-4}$ $(1.10^{-4})$ |
| EBM | $0.029$ $(0.006)$ | $0.025$ $(0.002)$ | $0.024$ $(0.002)$ | $0.025$ $(0.002)$ |

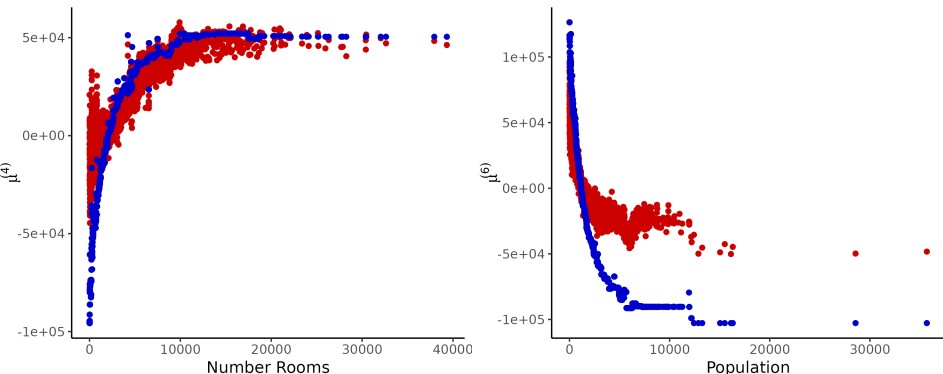

Figure 7: For the "Housing" dataset, main effects of the number of rooms and population in the house block in the decompositions of respectively TreeHFD in blue and TreeSHAP in red.

### B.3.1 Definition of performance metrics

**Accuracy.** The accuracy of `xgboost` models are the proportion of output explained variance, estimated by 10-fold cross-validation.

**Residual MSE.** The residual error of TreeHFD is defined as the MSE of TreeHFD predictions with respect to the tree ensemble predictions, normalized by the output variance $\mathbb{V}[T(\mathbf{X})]$. This ratio gives the magnitude of TreeHFD errors with respect to the original tree ensemble.

**Hierarchical orthogonality.** We evaluate the hierarchical orthogonality of the functional decompositions provided by TreeHFD and TreeSHAP, through the nine real data cases. For each dataset, we consider interaction components with a non-negligible influence in the decomposition, and thus select components with a variance of at least $1\%$ of the output variance $\mathbb{V}[T(\mathbf{X})]$. Then, we compute the correlation coefficients between the interaction components and their associated main effects, and report the maximum absolute value for each dataset in Table 2, for both TreeHFD and TreeSHAP. We only consider interactions with a non-null variance, since the correlation coefficient is undefined otherwise. When there is no interaction component above the threshold of $1\%$, we display "NA" in Table 2. We also recall that a functional decomposition is hierarchically orthogonal when all these correlations are null.

**Local variability.** We quantify the local variability of the two algorithms using the following procedure. We consider a given input variable, and for each observation of the dataset, we retrieve the ten nearest neighbors with respect to this variable, and compute the variance of the predictions from the associated component of the decomposition over these ten points. Then, the average variance over all observations of the dataset is derived, and normalized by the global variance of the functional component. Finally, this metric is averaged across all main effects. The obtained metric is reported in Table 2 in the "Local Variability" columns, and shows that TreeHFD is locally much more stable than TreeSHAP for all tested datasets. Notice that the metric is not very sensitive to the number of neighbors, and we get similar results using 5 or 20 instead of 10.

### B.3.2 Analysis of additional datasets

**Housing dataset.** For the "Housing" dataset presented in the article, we provide the additional Figure 7 for the main effects of the number of rooms and population in the house block, estimated by TreeHFD and TreeSHAP. We observe quite strong differences between the two algorithms.

**Superconductivity dataset.** The "Superconductivity" dataset (Kelly et al., 2024) is built from the extraction of 81 features from the chemical formula of 21263 superconductors, based on various characteristics, such as thermal conductivity, atomic radius, valence, electron affinity, and atomic mass. The output to be predicted is the superconducting critical temperature. We first fit a `xgboost` model with all default parameters and 100 trees, and then, TreeHFD is run with $d_I = 2$ to estimate the

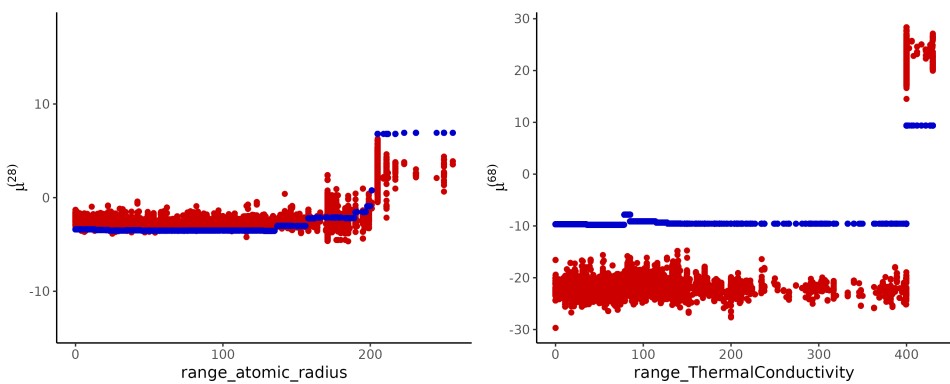

Figure 8: For the "Superconductivity" dataset, main effects of $X^{(28)}$ and $X^{(68)}$ in the decompositions of respectively TreeHFD in blue and TreeSHAP in red.

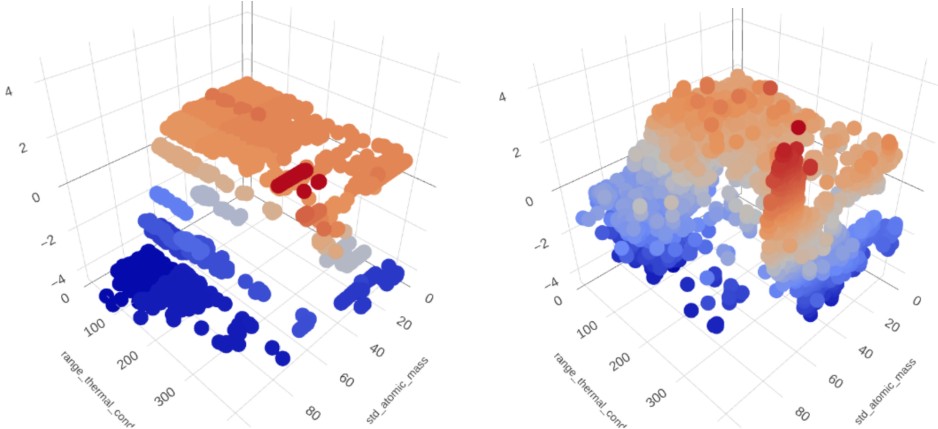

Figure 9: For the "Superconductivity" dataset, interaction between $X^{(10)}$ and $X^{(68)}$ in the decomposition, obtained with TreeHFD on the left, and TreeSHAP on the right.

HFD of the resulting ensemble. The `xgboost` model has a proportion of output explained variance of $92\%$ in this case, and the TreeHFD residual MSE is $1\%$ of $\mathbb{V}[T(\mathbf{X})]$. Then, we display the resulting decomposition in Figures 8 and 9. Again, we observe that TreeSHAP decomposition is quite noisy. On the other hand, TreeHFD exhibits clear functional relations for both main effects and interactions, although the input dimension of is quite large with $81$ variables. We also observe that TreeHFD and TreeSHAP provide quite different decompositions, and the targeted quantities are only clearly defined for TreeHFD.

Finally, we run a last experiment to illustrate that EBM and TreeHFD can also generate different decompositions, and EBM can be unstable. We randomly split the dataset in two halves, and compute both EBM and TreeHFD decompositions with each subsample. Figure 10 shows the resulting component for $X^{(68)}$, with EBM on the left and TreeHFD on the right. While the obtained TreeHFD component is stable across the two subsamples, it is not the case for EBM, which provides different functions.

**Bike Sharing dataset.** The "Bike sharing" dataset (Kelly et al., 2024) gathers data from a US bike renting system in 2011 and 2012. The number of bikes rented by hour is collected, along with time (hour, week day, holiday, season) and weather information (normalized feeling temperature, humidity, wind speed, weather conditions). We apply the same procedure than for the other datasets, and the proportion of output explained variance of the boosted tree ensemble is $87\%$, and the TreeHFD residual MSE is $3\%$ of the output variance $\mathbb{V}[T(\mathbf{X})]$. Finally, predictions are computed for all HFD components, and displayed in Figures 11 and 12, along with comparisons with the decomposition

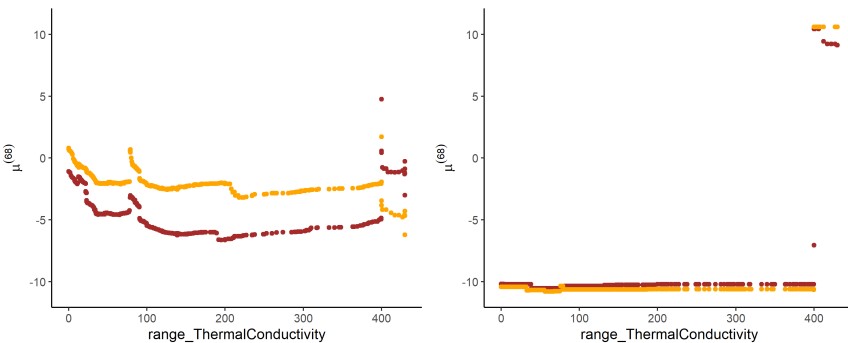

Figure 10: For the "Superconductivity" dataset, main effect of $X^{(68)}$ in the decompositions of respectively EBM on the left and TreeHFD on the right. The components obtained for the two subsamples are displayed in orange and brown on the same graph.

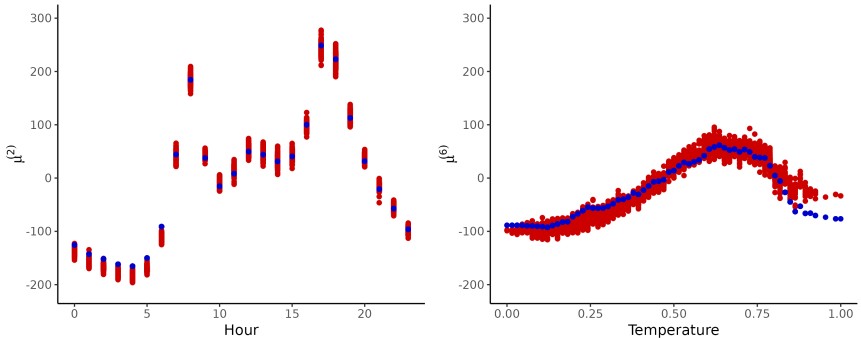

Figure 11: For the "Bike Sharing" dataset, main effects of "Hour" and "Temperature" in the decompositions of respectively TreeHFD in blue and TreeSHAP in red.

induced by TreeSHAP with interactions. We see that TreeHFD and TreeSHAP generate close decompositions, but SHAP values are quite noisy, and can therefore be misleading when predictions are analyzed one by one. Besides, the identified patterns can be easily interpreted. Indeed, there are renting peaks at the beginning and the end of the day, when people commute to work. Regarding the temperature, there is a drop of bike rentals when it is too hot or too cold, as expected. The main interaction term is between the variables "Hour" and "Week day", and is illustrated in Figure 12: During the weekend (days 0 and 6), the peak of bike rentals is in the middle of the day, instead of the morning and the evening for working days.

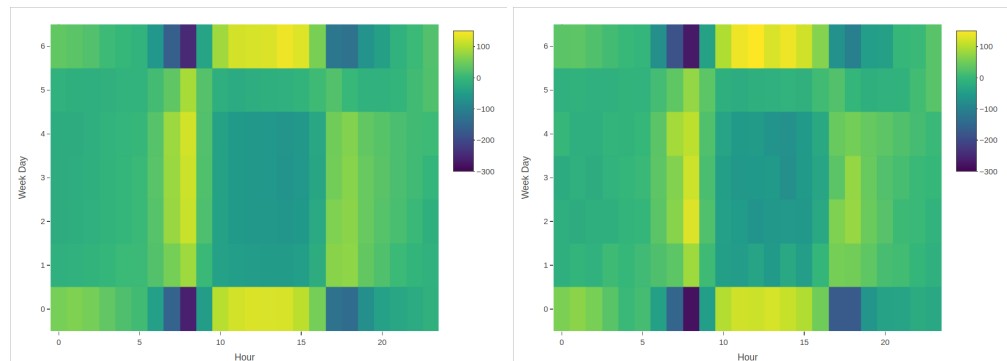

Figure 12: For the "Bike Sharing" dataset, interaction between "Hour" and "Week day", for TreeHFD on the left, and TreeSHAP on the right, both based on xgboost.

# C Proofs of theorems

## C.1 Proof of Theorem 2

For $J \in \mathcal{P}_p$, we denote by $f^{(J)}$ the marginal density of $\mathbf{X}^{(J)}$. For $\ell \in \mathcal{T}_M$, we also introduce the density $f_\ell^{(J)}$, defined as the density $f^{(J)}$ averaged over each cell of the partition $\mathcal{A}_\ell^{(J)}$, and then the random vector $\mathbf{Z}_\ell^{(J)}$ that admits density $f_\ell^{(J)}$. For $J = \mathcal{V}_p$, we simply write $\mathbf{Z}_\ell$ and $f_\ell$ to lighten notations.

The cornerstone of the proof of Theorem 2 is to apply the HFD of Theorem 1 to each tree $T_\ell(\mathbf{Z}_\ell)$, and to show that the discretized orthogonality constraints can also be written using $\mathbf{X}$.

**Lemma 1** (Tree HFD). *If Assumption 1 is satisfied, then for $\ell \in \mathcal{T}_M$, the tree prediction function $T_\ell(\mathbf{Z}_\ell)$ has a unique decomposition as the sum of the functions $\{\eta_\ell^{(J)}\}_{J \in \mathcal{P}_p}$, defined by*

$$T_\ell(\mathbf{Z}_\ell) = \sum_{J \in \mathcal{P}_p} \eta_\ell^{(J)}(\mathbf{Z}_\ell^{(J)}),$$

*and such that for all $J \in \mathcal{P}_p$, $I \subset J$ with $I \neq J$, $\mathbb{E}[\eta_\ell^{(J)}(\mathbf{Z}_\ell^{(J)})|\mathbf{Z}_\ell^{(I)}] = 0$. Additionally, for $J \in \mathcal{P}_p$, the function $\eta_\ell^{(J)}$ is piecewise constant over the partition $\mathcal{A}_\ell^{(J)}$.*

**Lemma 2.** *If Assumption 1 is satisfied, and for $\ell \in \mathcal{T}_M$ and $J \in \mathcal{P}_p$, $\mu_\ell^{(J)}$ is a function defined on $[0,1]^{|J|}$ and is piecewise constant on $\mathcal{A}_\ell^{(J)}$, then for $I \subset J$ such that $I \neq J$ and $A \in \mathcal{A}_\ell^{(I)}$, we have*

$$\mathbb{E}[\mu_\ell^{(J)}(\mathbf{X}^{(J)})|\mathbf{X}^{(I)} \in A] = \mathbb{E}[\mu_\ell^{(J)}(\mathbf{Z}_\ell^{(J)})|\mathbf{Z}_\ell^{(I)} \in A].$$

*Proof of Theorem 2.* Let Assumption 1 be satisfied. We proceed in two steps, to first show the existence of the decomposition, and then its uniqueness.

**Existence.** For each tree $\ell \in \mathcal{T}_M$, we apply Lemma 1 to get the HFD for $T_\ell(\mathbf{Z}_\ell)$, i.e., a unique decomposition as the sum of the piecewise constant functions $\{\eta_\ell^{(J)}\}_{J \in \mathcal{P}_p}$, defined by

$$T_\ell(\mathbf{Z}_\ell) = \sum_{J \in \mathcal{P}_p} \eta_\ell^{(J)}(\mathbf{Z}_\ell^{(J)}),$$

and such that for all $J \in \mathcal{P}_p$, $I \subset J$ with $I \neq J$, $\mathbb{E}[\eta_\ell^{(J)}(\mathbf{Z}_\ell^{(J)})|\mathbf{Z}_\ell^{(I)}] = 0$. Next, recall that the random vectors $\mathbf{Z}_\ell^{(J)}$ have the same distribution support $[0,1]^{|J|}$ as the vectors $\mathbf{X}^{(J)}$, and consequently $\{\eta_\ell^{(J)}(\mathbf{X}^{(J)})\}_J$ are well-defined. Then, according to Lemma 2, the orthogonality constraints implies that, for $I \subset J$ with $I \neq J$ and $A \in \mathcal{A}_\ell^{(I)}$,

$$\mathbb{E}[\eta_\ell^{(J)}(\mathbf{X}^{(J)})|\mathbf{X}^{(I)} \in A] = 0.$$

Therefore, we obtain the existence of the decomposition

$$T(\mathbf{X}) = \sum_{J \in \mathcal{P}_p} \eta^{(J)}(\mathbf{X}^{(J)}),$$

$$\text{with} \quad \eta^{(J)}(\mathbf{X}^{(J)}) = \sum_{\ell=1}^{M} \eta_\ell^{(J)}(\mathbf{X}^{(J)}), \quad T_\ell(\mathbf{X}) = \sum_{J \in \mathcal{P}_p} \eta_\ell^{(J)}(\mathbf{X}^{(J)}),$$

and such that for all $\ell \in \mathcal{T}_M$, $J \in \mathcal{P}_p$, $I \subset J$ with $I \neq J$, $A \in \mathcal{A}_\ell^{(I)}$, $\mathbb{E}[\eta_\ell^{(J)}(\mathbf{X}^{(J)})|\mathbf{X}^{(I)} \in A] = 0$, and where $\{\eta_\ell^{(J)}(\mathbf{X}^{(J)})\}$ are piecewise constant on the associated partitions.

**Uniqueness.** Notice that if the functions $\{\eta_\ell^{(J)}\}_J$ are not required to be piecewise constant, the decomposition is not unique in the general case, because the orthogonality constraints are weakened compared to the original HFD formulation. Indeed, we can directly apply Theorem 1 to the function $T(\mathbf{X})$ to get another decomposition, where the resulting components may not be piecewise constant, depending on the variations of the density $f$.

On the other hand, if the functions $\{\eta_\ell^{(J)}\}$ satisfy the decomposition and are piecewise constant, then according to Lemma 1, the functions $\{\eta_\ell^{(J)}\}$ are also the HFD of the trees for the vectors $\mathbf{Z}_\ell^{(J)}$, where the orthogonality constraints are transformed thanks to Lemma 2. Since each HFD tree decomposition is unique, the target global decomposition is unique.

$\square$

*Proof of Lemma 1.* We consider a given tree $\ell \in \mathcal{T}_M$. Since, the tree prediction function $T_\ell$ is piecewise constant on $[0,1]^p$, $T_\ell$ is square-integrable. Then, we can directly apply Theorem 1 from Chastaing et al. (2012) to the function $T_\ell(\mathbf{Z}_\ell)$ to get the existence of a unique set of functions $\{\eta_\ell^{(J)}\}_{J \in \mathcal{P}_p}$, such that the Hoeffding decomposition holds, i.e.,

$$T_\ell(\mathbf{Z}_\ell) = \sum_{J \in \mathcal{P}_p} \eta_\ell^{(J)}(\mathbf{Z}_\ell^{(J)}), \tag{1}$$

and for all $J \in \mathcal{P}_p$, $I \subset J$ with $I \neq J$, $\mathbb{E}[\eta_\ell^{(J)}(\mathbf{Z}_\ell^{(J)})|\mathbf{Z}_\ell^{(I)}] = 0$.

Now, we show that the functions $\eta_\ell^{(J)}$ are constant over $\mathcal{A}_\ell^{(J)}$. Let $A$ be a cell of $\mathcal{A}_\ell$. Using the HFD above, we have

$$\mathbb{E}[T_\ell(\mathbf{Z}_\ell)|\mathbf{Z}_\ell \in A] = \sum_{J \in \mathcal{P}_p} \mathbb{E}[\eta_\ell^{(J)}(\mathbf{Z}_\ell^{(J)})|\mathbf{Z}_\ell \in A].$$

By definition $T_\ell$ is constant over the cell $A$, and therefore

$$\mathbb{E}[T_\ell(\mathbf{Z}_\ell)|\mathbf{Z}_\ell \in A] = T_\ell(\mathbf{x}),$$

with $\mathbf{x} \in A$. On the other hand, $\mathbf{Z}_\ell$ has density $f_\ell$, which is constant over A. Consequently, the density of $\mathbf{Z}_\ell$ conditional on $\mathbf{Z}_\ell \in A$ takes the value $1/\mathrm{Vol}[A]$, where the operator $\mathrm{Vol}[.]$ gives the volume of a given cell. Then, we have for $J \in \mathcal{P}_p$,

$$\mathbb{E}[\eta_\ell^{(J)}(\mathbf{Z}_\ell^{(J)})|\mathbf{Z}_\ell \in A] = \int_A \eta_\ell^{(J)}(\mathbf{x}^{(J)}) \frac{1}{\mathrm{Vol}[A]} d\mathbf{x} = \frac{\mathrm{Vol}[A^{(-J)}]}{\mathrm{Vol}[A]} \int_{A^{(J)}} \eta_\ell^{(J)}(\mathbf{x}^{(J)}) d\mathbf{x}^{(J)}$$

$$= \frac{1}{\mathrm{Vol}[A^{(J)}]} \int_{A^{(J)}} \eta_\ell^{(J)}(\mathbf{x}^{(J)}) d\mathbf{x}^{(J)} = \mathbb{E}[\eta_\ell^{(J)}(\mathbf{Z}_\ell^{(J)})|\mathbf{Z}_\ell^{(J)} \in A^{(J)}].$$

Then, we have another decomposition of $T_\ell$ as the sum of the functions $\mu_\ell^{(J)}$, defined as

$$\mu_\ell^{(J)}(\mathbf{x}^{(J)}) = \sum_{A \in \mathcal{A}_\ell^{(J)}} \mathbb{E}[\eta_\ell^{(J)}(\mathbf{Z}_\ell^{(J)})|\mathbf{Z}_\ell^{(J)} \in A] \mathbb{1}_{\mathbf{x}^{(J)} \in A}.$$

Indeed, combining the above equations, we have

$$T_\ell(\mathbf{x}) = \sum_{J \in \mathcal{P}_p} \mu_\ell^{(J)}(\mathbf{x}^{(J)}).$$

Next, we can show that for all $I \subset J$ with $I \neq J$, $\mathbb{E}[\mu_\ell^{(J)}(\mathbf{Z}_\ell^{(J)})|\mathbf{Z}_\ell^{(J)}] = 0$, since

$$\mathbb{E}[\mu_\ell^{(J)}(\mathbf{Z}_\ell^{(J)})|\mathbf{Z}_\ell^{(I)}] = \sum_{A \in \mathcal{A}_\ell^{(J)}} \mathbb{E}[\eta_\ell^{(J)}(\mathbf{Z}_\ell^{(J)})|\mathbf{Z}_\ell^{(J)} \in A] \mathbb{P}(\mathbf{Z}_\ell^{(J)} \in A|\mathbf{Z}_\ell^{(I)})$$

$$= \mathbb{E}[\eta_\ell^{(J)}(\mathbf{Z}_\ell^{(J)})|\mathbf{Z}_\ell^{(I)}] = 0,$$

by definition of the HFD. Finally, since this decomposition is unique, we have for all $J \in \mathcal{P}_p$, $\eta_\ell^{(J)} = \mu_\ell^{(J)}$, and therefore, the functions $\eta_\ell^{(J)}$ are piecewise constant over $\mathcal{A}_\ell^{(J)}$. $\square$

*Proof of Lemma 2.* Let Assumption 1 be satisfied. For $\ell \in \mathcal{T}_M$ and $J \in \mathcal{P}_p$, we consider a square-integrable function $\mu_\ell^{(J)}$ defined on $[0,1]^p$ and piecewise constant on $\mathcal{A}_\ell^{(J)}$. With $f^{(J)}$ the marginal distribution of $\mathbf{X}^{(J)}$, we can expand

$$\mathbb{E}[\mu_\ell^{(J)}(\mathbf{X}^{(J)})|\mathbf{X}^{(I)} \in A^{(I)}] = \frac{1}{\mathbb{P}(\mathbf{X}^{(I)} \in A^{(I)})} \mathbb{E}[\mu_\ell^{(J)}(\mathbf{X}^{(J)}) \mathbb{1}_{\mathbf{X}^{(I)} \in A^{(I)}}]$$

$$= \frac{1}{\mathbb{P}(\mathbf{X}^{(I)} \in A^{(I)})} \int_{[0,1]^{|J|-|I|} \times A^{(I)}} \mu_\ell^{(J)}(\mathbf{x}^{(J)}) f^{(J)}(\mathbf{x}^{(J)}) d\mathbf{x}^{(J)}$$

$$= \frac{1}{\mathbb{P}(\mathbf{X}^{(I)} \in A^{(I)})} \sum_{A \in \mathcal{A}_\ell^{(J \setminus I)}} \int_{A \times A^{(I)}} \mu_\ell^{(J)}(\mathbf{x}^{(J)}) f^{(J)}(\mathbf{x}^{(J)}) d\mathbf{x}^{(J)}.$$

We introduce a list of vector points $\mathbf{x}_A^{(J)} \in A \times A^{(I)}$, for $A \in \mathcal{A}_\ell^{(J \setminus I)}$. Since $\mu_\ell^{(J)}$ is piecewise constant over the partition $\mathcal{A}_\ell^{(J)}$, we can write

$$\mathbb{E}[\mu_\ell^{(J)}(\mathbf{X}^{(J)})|\mathbf{X}^{(I)} \in A^{(I)}] = \frac{1}{\mathbb{P}(\mathbf{X}^{(I)} \in A^{(I)})} \sum_{A \in \mathcal{A}_\ell^{(J \setminus I)}} \mu_\ell^{(J)}(\mathbf{x}_A^{(J)}) \int_{A \times A^{(I)}} f^{(J)}(\mathbf{x}^{(J)}) d\mathbf{x}^{(J)}$$

$$= \frac{1}{\mathbb{P}(\mathbf{X}^{(I)} \in A^{(I)})} \sum_{A \in \mathcal{A}_\ell^{(J \setminus I)}} \mu_\ell^{(J)}(\mathbf{x}_A^{(J)}) f_\ell^{(J)}(\mathbf{x}_A^{(J)}) \mathrm{Vol}[A \times A^{(I)}],$$

where the last equality follows from the definition of $f_\ell^{(J)}$ as the average of $f^{(J)}$ over each cell of the partition. Similarly, $\mathbb{P}(\mathbf{X}^{(I)} \in A^{(I)}) = \mathbb{P}(\mathbf{Z}_\ell^{(I)} \in A^{(I)})$, and we get

$$\mathbb{E}[\mu_\ell^{(J)}(\mathbf{X}^{(J)})|\mathbf{X}^{(I)} \in A^{(I)}] = \frac{1}{\mathbb{P}(\mathbf{Z}_\ell^{(I)} \in A^{(I)})} \sum_{A \in \mathcal{A}_\ell^{(J \setminus I)}} \mu_\ell^{(J)}(\mathbf{x}_A^{(J)}) \mathbb{P}(\mathbf{Z}_\ell^{(J)} \in A \times A^{(I)})$$

$$= \sum_{A \in \mathcal{A}_\ell^{(J \setminus I)}} \mu_\ell^{(J)}(\mathbf{x}_A^{(J)}) \mathbb{P}(\mathbf{Z}_\ell^{(J)} \in A \times A^{(I)}|\mathbf{Z}_\ell^{(I)} \in A^{(I)})$$

$$= \mathbb{E}[\mu_\ell^{(J)}(\mathbf{Z}_\ell^{(J)})|\mathbf{Z}_\ell^{(I)} \in A^{(I)}],$$

which gives the final result. $\qquad\square$

### C.2 Proof of Theorem 3

*Proof of Theorem 3.* We apply Theorem 2 to get the tree HFD $\{\eta_\ell^{(J)}\}_{J \in \mathcal{P}_p, \ell \in \mathcal{T}_M}$ of $T(\mathbf{X})$. We consider a tree $\ell \in \mathcal{T}_M$, a variable set $J \in \mathcal{P}_p$, and an additional set $I \subset J$ with $I \neq J$.

For the first result of Theorem 3, we have

$$\mathrm{Cov}[\eta_\ell^{(J)}(\mathbf{X}^{(J)}), \eta_\ell^{(I)}(\mathbf{X}^{(I)})] = \mathbb{E}[\eta_\ell^{(J)}(\mathbf{X}^{(J)})\eta_\ell^{(I)}(\mathbf{X}^{(I)})]$$

$$= \sum_{A \in \mathcal{A}_\ell^{(I)}} \mathbb{E}[\eta_\ell^{(J)}(\mathbf{X}^{(J)})\eta_\ell^{(I)}(\mathbf{X}^{(I)})|\mathbf{X}^{(I)} \in A]\mathbb{P}(\mathbf{X}^{(I)} \in A).$$

Since $\eta_\ell^{(I)}$ is constant over each cell of $\mathcal{A}_\ell^{(I)}$ by construction, the above covariance writes

$$\mathrm{Cov}[\eta_\ell^{(J)}(\mathbf{X}^{(J)}), \eta_\ell^{(I)}(\mathbf{X}^{(I)})] = \sum_{A \in \mathcal{A}_\ell^{(I)}} \eta_\ell^{(I)}(A)\mathbb{E}[\eta_\ell^{(J)}(\mathbf{X}^{(J)})|\mathbf{X}^{(I)} \in A]\mathbb{P}(\mathbf{X}^{(I)} \in A),$$

with $\eta_\ell^{(I)}(A)$ the value of $\eta_\ell^{(I)}(\mathbf{x})$ for $\mathbf{x} \in A$. Finally, using the discretized orthogonality constraints, we conclude that $\eta_\ell^{(J)}$ and $\eta_\ell^{(I)}$ are orthogonal, i.e., $\mathrm{Cov}[\eta_\ell^{(J)}(\mathbf{X}^{(J)}), \eta_\ell^{(I)}(\mathbf{X}^{(I)})] = 0$.

For the second part of Theorem 3, we consider a given point $\mathbf{x}_0^{(I)} \in [0, 1]^{|I|}$. The cell of $\mathcal{A}_\ell^{(I)}$ containing $\mathbf{x}_0^{(I)}$ is denoted by $A^{(I)}$. Since $\eta_\ell^{(J)}$ is constant over each cell $A$ of $\mathcal{A}_\ell^{(J)}$, we write $\eta_\ell^{(J)}(A)$ this constant value.

Then, we can write

$$\mathbb{E}[\eta_\ell^{(J)}(\mathbf{X}^{(J)})|\mathbf{X}^{(I)} = \mathbf{x}_0^{(I)}] = \sum_{A \in \mathcal{A}_\ell^{(J \setminus I)}} \eta_\ell^{(J)}(A \times A^{(I)})\mathbb{P}(\mathbf{X}^{(J \setminus I)} \in A|\mathbf{X}^{(I)} = \mathbf{x}_0^{(I)}),$$

$$\mathbb{E}[\eta_\ell^{(J)}(\mathbf{X}^{(J)})|\mathbf{X}^{(I)} \in A^{(I)}] = \sum_{A \in \mathcal{A}_\ell^{(J \setminus I)}} \eta_\ell^{(J)}(A \times A^{(I)})\mathbb{P}(\mathbf{X}^{(J \setminus I)} \in A|\mathbf{X}^{(I)} \in A^{(I)}).$$

According to Theorem 2, the second equation is null, since $\eta_\ell^{(J)}$ satisfy the hierarchical orthogonality constraints. Therefore, we can write

$$\mathbb{E}[\eta_\ell^{(J)}(\mathbf{X}^{(J)})|\mathbf{X}^{(I)} = \mathbf{x}_0^{(I)}] = \sum_{A \in \mathcal{A}_\ell^{(J \setminus I)}} \eta_\ell^{(J)}(A \times A^{(I)})\big[\mathbb{P}(\mathbf{X}^{(J \setminus I)} \in A|\mathbf{X}^{(I)} = \mathbf{x}_0^{(I)})$$

$$- \mathbb{P}(\mathbf{X}^{(J \setminus I)} \in A|\mathbf{X}^{(I)} \in A^{(I)})\big],$$

and get

$$\mathbb{E}[\eta_\ell^{(J)}(\mathbf{X}^{(J)})|\mathbf{X}^{(I)} = \mathbf{x}_0^{(I)}] = \sum_{A \in \mathcal{A}_\ell^{(J \setminus I)}} \frac{\eta_\ell^{(J)}(A \times A^{(I)})}{\mathbb{P}(\mathbf{X}^{(I)} \in A^{(I)})} \big[ \mathbb{P}(\mathbf{X}^{(J \setminus I)} \in A | \mathbf{X}^{(I)} = \mathbf{x}_0^{(I)}) \mathbb{P}(\mathbf{X}^{(I)} \in A^{(I)})$$
$$- \mathbb{P}(\mathbf{X}^{(J)} \in A \times A^{(I)}) \big]. \qquad (2)$$

Next, we expand the involved probabilities using the density $f^{(J)}$ of the vector $\mathbf{X}^{(J)}$ to obtain

$$\mathbb{P}(\mathbf{X}^{(J)} \in A \times A^{(I)}) = \int_{A \times A^{(I)}} f^{(J)}(\mathbf{x}^{(J)}) d\mathbf{x}^{(J)},$$

$$\mathbb{P}(\mathbf{X}^{(I)} \in A^{(I)}) = \int_{A^{(I)} \times [0,1]^{|J \setminus I|}} f^{(J)}(\mathbf{x}^{(J)}) d\mathbf{x}^{(J)},$$

$$\mathbb{P}(\mathbf{X}^{(J \setminus I)} \in A | \mathbf{X}^{(I)} = \mathbf{x}_0^{(I)}) = \frac{\int_A f^{(J)}(\mathbf{x}^{(J \setminus I)}, \mathbf{x}_0^{(I)}) d\mathbf{x}^{(J \setminus I)}}{\int_{[0,1]^{|J \setminus I|}} f^{(J)}(\mathbf{x}^{(J \setminus I)}, \mathbf{x}_0^{(I)}) d\mathbf{x}^{(J \setminus I)}}.$$

We recall that the maximum variability of the density $f$ over a cell of the tree partitions is defined by

$$\Delta_{\mathcal{A}, f} = \sup_{A \in \cup_\ell \mathcal{A}_\ell} \sup_{\mathbf{x}, \mathbf{x}' \in A} |f(\mathbf{x}) - f(\mathbf{x}')|.$$

Since $f^{(J)}$ is a marginal density of $f$, we have $\Delta_{\mathcal{A}, f^{(J)}} \leq \Delta_{\mathcal{A}, f}$. Additionally, $f$ is bounded by positive constants $c_1, c_2 > 0$ according to Assumption 1, i.e $c_1 \leq f^{(J)}(\mathbf{x}^{(J)}) \leq c_2$. Then, we can bound the above integrals as follows. First,

$$\frac{1}{\int_{[0,1]^{|J \setminus I|}} f^{(J)}(\mathbf{x}^{(J \setminus I)}, \mathbf{x}_0^{(I)}) d\mathbf{x}^{(J \setminus I)}} \leq 1/c_1,$$

and

$$\mathbb{P}(\mathbf{X}^{(I)} \in A^{(I)}) = \int_{A^{(I)} \times [0,1]^{|J \setminus I|}} f^{(J)}(\mathbf{x}^{(J)}) d\mathbf{x}^{(J)} \leq c_2 \mathrm{Vol}(A^{(I)}).$$

Then, we separate the variables $\mathbf{x}^{(J \setminus I)}$ and $\mathbf{x}^{(I)}$ in the integral of the right hand side of the following line to get

$$\mathrm{Vol}(A^{(I)}) \int_A f^{(J)}(\mathbf{x}^{(J \setminus I)}, \mathbf{x}_0^{(I)}) d\mathbf{x}^{(J \setminus I)} = \int_{A \times A^{(I)}} f^{(J)}(\mathbf{x}^{(J \setminus I)}, \mathbf{x}_0^{(I)}) d\mathbf{x}^{(J)}$$

$$\leq \int_{A \times A^{(I)}} f^{(J)}(\mathbf{x}^{(J)}) d\mathbf{x}^{(J)} + \int_{A \times A^{(I)}} |f^{(J)}(\mathbf{x}^{(J \setminus I)}, \mathbf{x}_0^{(I)}) - f^{(J)}(\mathbf{x}^{(J)})| d\mathbf{x}^{(J)}$$

$$\leq \int_{A \times A^{(I)}} f^{(J)}(\mathbf{x}^{(J)}) d\mathbf{x}^{(J)} + \Delta_{\mathcal{A}, f} \mathrm{Vol}(A \times A^{(I)})$$

$$\leq \mathbb{P}(\mathbf{X}^{(J)} \in A \times A^{(I)}) + \Delta_{\mathcal{A}, f} \mathrm{Vol}(A \times A^{(I)}).$$

Similarly, we also have

$$\mathrm{Vol}(A^{(I)}) \int_{[0,1]^{|J \setminus I|}} f^{(J)}(\mathbf{x}^{(J \setminus I)}, \mathbf{x}_0^{(I)}) d\mathbf{x}^{(J \setminus I)} = \sum_{A \in \mathcal{A}_\ell^{(J \setminus I)}} \int_{A \times A^{(I)}} f^{(J)}(\mathbf{x}^{(J \setminus I)}, \mathbf{x}_0^{(I)}) d\mathbf{x}^{(J)}$$

$$\leq \sum_{A \in \mathcal{A}_\ell^{(J \setminus I)}} \int_{A \times A^{(I)}} f^{(J)}(\mathbf{x}^{(J)}) d\mathbf{x}^{(J)} + \Delta_{\mathcal{A}, f} \mathrm{Vol}(A \times A^{(I)})$$

$$\leq \int_{A^{(I)} \times [0,1]^{|J \setminus I|}} f^{(J)}(\mathbf{x}^{(J)}) d\mathbf{x}^{(J)} + \Delta_{\mathcal{A}, f} \mathrm{Vol}(A^{(I)})$$

$$\leq \mathbb{P}(\mathbf{X}^{(I)} \in A^{(I)}) + \Delta_{\mathcal{A}, f} \mathrm{Vol}(A^{(I)}).$$

Overall, we obtain

$$
\left| \mathbb{P}(\mathbf{X}^{(J\setminus I)} \in A | \mathbf{X}^{(I)} = \mathbf{x}_0^{(I)}) \mathbb{P}(\mathbf{X}^{(I)} \in A^{(I)}) - \mathbb{P}(\mathbf{X}^{(J)} \in A \times A^{(I)}) \right|
$$

$$
= \frac{1}{\int_{[0,1]^{|J\setminus I|}} f^{(J)}(\mathbf{x}^{(J\setminus I)}, \mathbf{x}_0^{(I)}) d\mathbf{x}^{(J\setminus I)}} \left| \int_A f^{(J)}(\mathbf{x}^{(J\setminus I)}, \mathbf{x}_0^{(I)}) d\mathbf{x}^{(J\setminus I)} \mathbb{P}(\mathbf{X}^{(I)} \in A^{(I)}) \right.
$$

$$
\left. - \mathbb{P}(\mathbf{X}^{(J)} \in A \times A^{(I)}) \int_{[0,1]^{|J\setminus I|}} f^{(J)}(\mathbf{x}^{(J\setminus I)}, \mathbf{x}_0^{(I)}) d\mathbf{x}^{(J\setminus I)} \right|
$$

$$
\leq \frac{1}{c_1 \mathrm{Vol}(A^{(I)})} \left| \left[ \mathbb{P}(\mathbf{X}^{(J)} \in A \times A^{(I)}) + \Delta_{\mathcal{A},f} \mathrm{Vol}(A \times A^{(I)}) \right] \mathbb{P}(\mathbf{X}^{(I)} \in A^{(I)}) \right.
$$

$$
\left. - \mathbb{P}(\mathbf{X}^{(J)} \in A \times A^{(I)}) \left[ \mathbb{P}(\mathbf{X}^{(I)} \in A^{(I)}) + \Delta_{\mathcal{A},f} \mathrm{Vol}(A^{(I)}) \right] \right|
$$

$$
\leq \frac{1}{c_1 \mathrm{Vol}(A^{(I)})} \left| \Delta_{\mathcal{A},f} \mathrm{Vol}(A \times A^{(I)}) \mathbb{P}(\mathbf{X}^{(I)} \in A^{(I)}) - \mathbb{P}(\mathbf{X}^{(J)} \in A \times A^{(I)}) \Delta_{\mathcal{A},f} \mathrm{Vol}(A^{(I)}) \right|
$$

$$
\leq \frac{\Delta_{\mathcal{A},f}}{c_1} \left| \mathrm{Vol}(A) \mathbb{P}(\mathbf{X}^{(I)} \in A^{(I)}) - \mathbb{P}(\mathbf{X}^{(J)} \in A \times A^{(I)}) \right|
$$

$$
\leq \frac{c_2 - c_1}{c_1} \mathrm{Vol}(A \times A^{(I)}) \Delta_{\mathcal{A},f},
$$

where the last step follows from $c_1 \mathrm{Vol}(A^{(I)}) \leq \mathbb{P}(\mathbf{X}^{(I)} \in A^{(I)}) \leq c_2 \mathrm{Vol}(A^{(I)})$. Finally, using Equation (2),

$$
\mathbb{E}[\eta_\ell^{(J)}(\mathbf{X}^{(J)}) | \mathbf{X}^{(I)} = \mathbf{x}_0^{(I)}] \leq \sum_{A \in \mathcal{A}_\ell^{(J\setminus I)}} \frac{|\eta_\ell^{(J)}(A \times A^{(I)})|}{c_1 \mathrm{Vol}(A^{(I)})} \frac{c_2 - c_1}{c_1} \mathrm{Vol}(A \times A^{(I)}) \Delta_{\mathcal{A},f}
$$

$$
\leq \Delta_{\mathcal{A},f} \frac{c_2 - c_1}{c_1^2} \sum_{A \in \mathcal{A}_\ell^{(J\setminus I)}} |\eta_\ell^{(J)}(A \times A^{(I)})| \mathrm{Vol}(A)
$$

$$
\leq \Delta_{\mathcal{A},f} \frac{c_2 - c_1}{c_1^2} \mathbb{E}[|\eta_\ell^{(J)}(\mathbf{U})| \mid \mathbf{U}^{(I)} \in A^{(I)}],
$$

where $\mathbf{U}$ is a uniform variable on $[0,1]^{|J|}$. Therefore, we have

$$
\left| \mathbb{E}[\eta_\ell^{(J)}(\mathbf{X}^{(J)}) | \mathbf{X}^{(I)}] \right| \leq \frac{c_2 - c_1}{c_1^2} \|\eta_\ell^{(J)}\|_\infty \Delta_{\mathcal{A},f}.
$$

We now conclude the proof by the analysis of the targeted covariance using Jensen's inequality, the triangle inequality, and the above inequality,

$$
\left| \mathrm{Cov}[\eta^{(J)}(\mathbf{X}^{(J)}), \eta^{(I)}(\mathbf{X}^{(I)})] \right| = \left| \mathbb{E}[\eta^{(J)}(\mathbf{X}^{(J)}) \eta^{(I)}(\mathbf{X}^{(I)})] \right| = \left| \mathbb{E}[\mathbb{E}[\eta^{(J)}(\mathbf{X}^{(J)}) \eta^{(I)}(\mathbf{X}^{(I)}) | \mathbf{X}^{(I)}]] \right|
$$

$$
= \left| \mathbb{E}[\eta^{(I)}(\mathbf{X}^{(I)}) \mathbb{E}[\eta^{(J)}(\mathbf{X}^{(J)}) | \mathbf{X}^{(I)}]] \right|
$$

$$
\leq \|\eta^{(I)}\|_\infty \sum_{\ell=1}^M \left| \mathbb{E}[\eta_\ell^{(J)}(\mathbf{X}^{(J)}) | \mathbf{X}^{(I)}] \right|
$$

$$
\leq \frac{c_2 - c_1}{c_1^2} \|\eta^{(I)}\|_\infty \left( \sum_{\ell=1}^M \|\eta_\ell^{(J)}\|_\infty \right) \Delta_{\mathcal{A},f},
$$

which concludes the proof. $\qquad\square$

### C.3 Proof of Theorem 4

*Proof of Theorem 4.* We consider a pair of variable sets $I, I' \in \mathcal{P}_p$ and the prediction function of a tree ensemble $T = \sum_\ell T_\ell$, such that $T_\ell$ has a decomposition $T_\ell(\mathbf{X}) = g_\ell(\mathbf{X}^{(I)}) + h_\ell(\mathbf{X}^{(I')})$ for a pair of functions $g_\ell, h_\ell$. Since $T_\ell$ is piecewise constant over $\mathcal{A}_\ell$, we can average the functions $g_\ell$ and $h_\ell$ over each cell of $\mathcal{A}_\ell$, and the decomposition of $T_\ell$ still holds. Therefore, we have

$T_\ell(\mathbf{X}) = g_\ell(\mathbf{X}^{(I)}) + h_\ell(\mathbf{X}^{(I')})$, with $g_\ell$ and $h_\ell$ two piecewise constant functions over $\mathcal{A}_\ell$. Now, we can apply the tree HFD of Theorem 2 to the functions $g_\ell$ and $h_\ell$, to get

$$g_\ell(\mathbf{X}^{(I)}) = \sum_{J \subset I} \eta_\ell^{(J)}(\mathbf{X}^{(J)}),$$

$$h_\ell(\mathbf{X}^{(I)}) = \sum_{J \subset I'} \eta_\ell'^{(J)}(\mathbf{X}^{(J)}),$$

where $\{\eta_\ell^{(J)}\}_{J \subset I}$ and $\{\eta_\ell'^{(J)}\}_{J \subset I'}$ are piecewise constant functions over the Cartesian tree partitions of $T$, and satisfy the orthogonality constraints. By construction, we have

$$T_\ell(\mathbf{X}) = \sum_{J \subset I} \eta_\ell^{(J)}(\mathbf{X}^{(J)}) + \sum_{J \subset I'} \eta_\ell'^{(J)}(\mathbf{X}^{(J)}),$$

which defines the tree HFD of $T_\ell(\mathbf{X})$ using the uniqueness property of the tree HFD. In particular, for $J \subset I \cap I'$, the tree HFD component associated with $J$ is given by $\eta_\ell^{(J)} + \eta_\ell'^{(J)}$. Then, we conclude that for all $J \in \mathcal{P}_p$ such that $J \not\subset I$ and $J \not\subset I'$, the tree HFD component $\eta^{(J)}$ of $T$ satisfies almost surely

$$\eta^{(J)}(\mathbf{X}^{(J)}) = 0.$$

$\square$

## C.4 Proof of Theorem 5

*Proof of Theorem 5.* Let $\ell \in \mathcal{T}_M$, $\{\eta_\ell^{(J)}\}_{J \in \mathcal{P}_p}$ be the tree HFD defined in Theorem 2, and $\{\phi_\ell^{(J)}\}_{J \in \mathcal{P}_p}$ the Shapley-GAM decomposition induced by the interventional SHAP value function $v^{(int)}$ (Bordt and von Luxburg, 2023), defined by the marginal expectation as

$$v^{(int)}(J, \mathbf{x}^{(J)}) = \mathbb{E}[T_\ell(\mathbf{x}^{(J)}, \mathbf{X}^{(-J)})].$$

Then, we can define the sets of exogenous variables for the two functional decompositions as

$$\Gamma_h = \{j \in \mathcal{V}_p : \forall J \in \mathcal{P}_p, \eta_\ell^{(J \cup \{j\})} = 0\}$$

$$\Gamma_s = \{j \in \mathcal{V}_p : \forall J \in \mathcal{P}_p, \phi_\ell^{(J \cup \{j\})} = 0\}.$$

By definition, we have

$$T_\ell(\mathbf{X}) = \sum_{J \subset \mathcal{V}_p \setminus \Gamma_s} \phi_\ell^{(J)}(\mathbf{X}^{(J)}).$$

We directly apply Theorem 4 to get that for $J \not\subset \mathcal{V}_p \setminus \Gamma_s$,

$$\eta_\ell^{(J)} = 0.$$

Therefore, we have $\Gamma_s \subset \Gamma_h$.

Alternatively, we also have

$$T_\ell(\mathbf{X}) = \sum_{J \subset \mathcal{V}_p \setminus \Gamma_h} \eta_\ell^{(J)}(\mathbf{X}^{(J)}).$$

First, notice that the interventional value function writes, for $A \in \mathcal{P}_p$,

$$v^{(int)}(A, \mathbf{x}^{(A)}) = \mathbb{E}[T_\ell(\mathbf{x}^{(A)}, \mathbf{X}^{(-A)})] = \sum_{J \subset \mathcal{V}_p \setminus \Gamma_h} \mathbb{E}[\eta_\ell^{(J)}(\mathbf{x}^{(J \cap A)}, \mathbf{X}^{(J \setminus A)})].$$

Then, we consider $j \in \Gamma_h$, and $I \subset \mathcal{V}_p$ such that $j \in I$, and apply Theorem 4 from Bordt and von Luxburg (2023) to get

$$\phi_\ell^{(I)}(\mathbf{x}^{(I)}) = \sum_{A \subset I} (-1)^{|I| - |A|} v^{(int)}(A, \mathbf{x}^{(A)})$$

$$= \sum_{A \subset I \setminus \{j\}} (-1)^{|I| - |A|} v^{(int)}(A, \mathbf{x}^{(A)})$$

$$+ \sum_{A \subset I \setminus \{j\}} (-1)^{|I| - |A \cup \{j\}|} \sum_{J \subset \mathcal{V}_p \setminus \Gamma_h} \mathbb{E}[\eta_\ell^{(J)}(\mathbf{x}^{(J \cap (A \cup \{j\}))}, \mathbf{X}^{(J \setminus (A \cup \{j\}))})].$$

In the last sum of the above equation, we have $j \in \Gamma_h$ and $J \subset \mathcal{V}_p \setminus \Gamma_h$. Therefore, $J \cap (A \cup \{j\}) = J \cap A$ and $J \setminus (A \cup \{j\}) = J \setminus A$. Hence,

$$
\begin{aligned}
\phi_\ell^{(I)}(\mathbf{x}^{(I)}) &= \sum_{A \subset I \setminus \{j\}} \left[ (-1)^{|I|-|A|} + (-1)^{|I|-|A \cup \{j\}|} \right] v^{(int)}(A, \mathbf{x}^{(A)}) \\
&= \sum_{A \subset I \setminus \{j\}} \left[ (-1)^{|I|-|A|} - (-1)^{|I|-|A|} \right] v^{(int)}(A, \mathbf{x}^{(A)}) \\
&= 0.
\end{aligned}
$$

Consequently, $j \in \Gamma_s$, and then $\Gamma_h \subset \Gamma_s$. Overall, we obtain $\Gamma_h = \Gamma_s$. $\qquad\square$

### C.5  Proof of Theorem 6

We first recall the bounds of the distribution $f$ of $\mathbf{X}$ in Assumption 1: there exists $c_1, c_2 > 0$ such that for all $\mathbf{x} \in [0,1]^p$, $c_1 \le f(\mathbf{x}) \le c_2$. For $J \in \mathcal{P}_p$, and $I \subset J$ with $I \ne J$, we also denote by $f^{(J|I)}(\mathbf{x})$ the distribution of $\mathbf{X}^{(J)}$ conditional on $\mathbf{X}^{(I)} = \mathbf{x}^{(I)}$, and use the same notation for all conditional distributions.

**Lemma 3** (Lemma 3.1 from (Stone, 1994)). *If Assumption 1 is satisfied, $\nu$ is a square-integrable function defined on $[0,1]^p$, and $\{\nu^{(J)}\}_J$ is the original HFD of $\nu$, then we have*

$$
\mathbb{E}[\nu(\mathbf{X})^2] \ge \left( 1 - \sqrt{1 - \frac{c_1}{c_2^2}} \right)^{2^p - 1} \sum_{J \in \mathcal{P}_p} \mathbb{E}[\nu^{(J)}(\mathbf{X}^{(J)})^2].
$$

**Lemma 4.** *If Assumption 1 is satisfied, $\{T_\ell^{(J)}\}_J$ is the original HFD of $T_\ell(\mathbf{X})$, and $\{\eta_\ell^{(J)}\}_J$ is the tree HFD of $T_\ell(\mathbf{X})$, then we have for $J \in \mathcal{P}_p$,*

$$
\eta_\ell^{(J)} = T_\ell^{(J)} \frac{f^{(J)}}{f_\ell^{(J)}} - \delta_\ell^{(J)},
$$

*where $\{\delta_\ell^{(J)}\}_J$ is the HFD of $\delta_\ell(\mathbf{Z}_\ell)$, with*

$$
\delta_\ell = \sum_{J \in \mathcal{P}_p} T_\ell^{(J)} \left( \frac{f^{(J)}}{f_\ell^{(J)}} - 1 \right).
$$

**Lemma 5.** *If Assumption 1 is satisfied, then there exists a constant $K > 0$ such that for any tree $T_\ell$, we have*

$$
\sum_{J \in \mathcal{P}_p} \mathbb{E}[(\eta_\ell^{(J)}(\mathbf{X}^{(J)}) - T_\ell^{(J)}(\mathbf{X}^{(J)}))^2] \le K \Delta_{\mathcal{A}, f}^2 \mathbb{E}[T_\ell(\mathbf{X})^2],
$$

*where $\{T_\ell^{(J)}\}_J$ is the original HFD of $T_\ell(\mathbf{X})$, and $\{\eta_\ell^{(J)}\}_J$ is the tree HFD of $T_\ell(\mathbf{X})$.*

*Proof of Theorem 6.* We first split the MSE of interest using the original HFD $\{T^{(J)}\}_J$ of $T(\mathbf{X})$, as follows

$$
\mathbb{E}[(\eta^{(J)}(\mathbf{X}^{(J)}) - m^{(J)}(\mathbf{X}^{(J)}))^2] = \mathbb{E}[([\eta^{(J)}(\mathbf{X}^{(J)}) - T^{(J)}(\mathbf{X}^{(J)})] + [T^{(J)}(\mathbf{X}^{(J)}) - m^{(J)}(\mathbf{X}^{(J)})])^2].
$$

Recall that for two real numbers $a, b$, we have $(a + b)^2 \le 2(a^2 + b^2)$. Hence, we have

$$
\begin{aligned}
\mathbb{E}[(\eta^{(J)}(\mathbf{X}^{(J)}) - m^{(J)}(\mathbf{X}^{(J)}))^2] \le\ &2\mathbb{E}[(\eta^{(J)}(\mathbf{X}^{(J)}) - T^{(J)}(\mathbf{X}^{(J)}))^2] \\
&+ 2\mathbb{E}[(T^{(J)}(\mathbf{X}^{(J)}) - m^{(J)}(\mathbf{X}^{(J)}))^2].
\end{aligned}
$$

The core of the proof is thus to bound the two terms of the right hand side of the above inequality. We first focus on $\mathbb{E}[(\eta^{(J)}(\mathbf{X}^{(J)}) - T^{(J)}(\mathbf{X}^{(J)}))^2]$. If we denote by $\{T_\ell^{(J)}\}_J$ the original HFD of each tree $T_\ell(\mathbf{X})$, the HFD uniqueness implies that

$$
T^{(J)}(\mathbf{X}^{(J)}) = \sum_{\ell=1}^{M} T_\ell^{(J)}(\mathbf{X}^{(J)}).
$$

Consequently,

$$\mathbb{E}[(\eta^{(J)}(\mathbf{X}^{(J)}) - T^{(J)}(\mathbf{X}^{(J)}))^2] = \mathbb{E}\Big[\Big(\sum_{\ell=1}^{M}\eta_\ell^{(J)}(\mathbf{X}^{(J)}) - T_\ell^{(J)}(\mathbf{X}^{(J)})\Big)^2\Big]$$

$$= \mathbb{E}\Big[\sum_{\ell,\ell'=1}^{M}[\eta_\ell^{(J)}(\mathbf{X}^{(J)}) - T_\ell^{(J)}(\mathbf{X}^{(J)})][\eta_{\ell'}^{(J)}(\mathbf{X}^{(J)}) - T_{\ell'}^{(J)}(\mathbf{X}^{(J)})]\Big]$$

$$\leq \sum_{\ell,\ell'=1}^{M}\sqrt{\mathbb{E}[(\eta_\ell^{(J)}(\mathbf{X}^{(J)}) - T_\ell^{(J)}(\mathbf{X}^{(J)}))^2]\mathbb{E}[(\eta_{\ell'}^{(J)}(\mathbf{X}^{(J)}) - T_{\ell'}^{(J)}(\mathbf{X}^{(J)}))^2]},$$

using Cauchy-Schwartz inequality at the last step. Finally, we apply Lemma 5 to obtain that there exists a constant $K_2 > 0$ such that

$$\mathbb{E}[(\eta^{(J)}(\mathbf{X}^{(J)}) - T^{(J)}(\mathbf{X}^{(J)}))^2] \leq \frac{K_2}{2}\Delta_{\mathcal{A},f}^2\Big(\sum_{\ell=1}^{M}\sqrt{\mathbb{E}[T_\ell(\mathbf{X})^2]}\Big)^2.$$

For the second term $\mathbb{E}[(T^{(J)}(\mathbf{X}^{(J)}) - m^{(J)}(\mathbf{X}^{(J)}))^2]$, we use again the HFD uniqueness to get that the original HFD of $T(\mathbf{X}) - m(\mathbf{X})$ is given by $\{T^{(J)}(\mathbf{X}^{(J)}) - m^{(J)}(\mathbf{X}^{(J)})\}_J$. We apply Lemma 3 to $T(\mathbf{X}) - m(\mathbf{X})$ to get that there exists a constant $K_1 > 0$ such that

$$\mathbb{E}[(T^{(J)}(\mathbf{X}^{(J)}) - m^{(J)}(\mathbf{X}^{(J)}))^2] \leq \frac{K_1}{2}\mathbb{E}[(m(\mathbf{X}) - T(\mathbf{X}))^2].$$

We combine the two last inequalities to obtain the final result, that is

$$\mathbb{E}[(\eta^{(J)}(\mathbf{X}^{(J)}) - m^{(J)}(\mathbf{X}^{(J)}))^2] \leq K_1\mathbb{E}[(m(\mathbf{X}) - T(\mathbf{X}))^2] + K_2\Delta_{\mathcal{A},f}^2\Big(\sum_{\ell=1}^{M}\sqrt{\mathbb{E}[T_\ell(\mathbf{X})^2]}\Big)^2.$$

$\square$

*Proof of Lemma 4.* By construction, we have

$$\sum_{J\in\mathcal{P}_p}\eta_\ell^{(J)} = T_\ell.$$

Therefore, we simply need to show that the orthogonality constraints of the tree HFD are satisfied by $\{\eta_\ell^{(J)}\}_J$ to obtain the final result using the uniqueness of the tree HFD. Hence, we write for $I \subset J$ with $I \neq J$,

$$\mathbb{E}[\eta_\ell^{(J)}(\mathbf{Z}_\ell^{(J)})|\mathbf{Z}_\ell^{(I)}] = \mathbb{E}\Big[T_\ell^{(J)}(\mathbf{Z}_\ell^{(J)})\frac{f^{(J)}(\mathbf{Z}_\ell^{(J)})}{f_\ell^{(J)}(\mathbf{Z}_\ell^{(J)})} - \delta_\ell^{(J)}(\mathbf{Z}_\ell^{(J)})|\mathbf{Z}_\ell^{(I)}\Big]$$

$$= \mathbb{E}\Big[T_\ell^{(J)}(\mathbf{Z}_\ell^{(J)})\frac{f^{(J)}(\mathbf{Z}_\ell^{(J)})}{f_\ell^{(J)}(\mathbf{Z}_\ell^{(J)})}|\mathbf{Z}_\ell^{(I)}\Big],$$

since $\delta_\ell^{(J)}(\mathbf{Z}_\ell^{(J)})$ satisfies the orthogonality constraints by definition. Next, recall that conditional on $\mathbf{Z}_\ell^{(I)}$, $\mathbf{Z}_\ell^{(J)}$ has density $f_\ell^{(J|I)}$, and $\mathbf{X}^{(J)}$ conditional on $\mathbf{X}^{(I)}$, has density $f^{(J|I)}$, which are both positive. Therefore, for $\mathbf{x}^{(I)} \in [0,1]^{|I|}$ we have

$$\mathbb{E}\Big[T_\ell^{(J)}(\mathbf{Z}_\ell^{(J)})\frac{f^{(J)}(\mathbf{Z}_\ell^{(J)})}{f_\ell^{(J)}(\mathbf{Z}_\ell^{(J)})}|\mathbf{Z}_\ell^{(I)} = \mathbf{x}^{(I)}\Big]$$

$$= \mathbb{E}\Big[T_\ell^{(J)}(\mathbf{X}^{(J)})\frac{f^{(J)}(\mathbf{X}^{(J)})}{f_\ell^{(J)}(\mathbf{X}^{(J)})}\frac{f_\ell^{(J|I)}(\mathbf{X}^{(J)})}{f^{(J|I)}(\mathbf{X}^{(J)})}|\mathbf{X}^{(I)} = \mathbf{x}^{(I)}\Big]$$

$$= \frac{\int f^{(J)}(\mathbf{x})d\mathbf{x}^{(J\backslash I)}}{\int f_\ell^{(J)}(\mathbf{x})d\mathbf{x}^{(J\backslash I)}}\mathbb{E}\Big[T_\ell^{(J)}(\mathbf{X}^{(J)})|\mathbf{X}^{(I)} = \mathbf{x}^{(I)}\Big] = 0,$$

since $T_\ell^{(J)}$ satisfies the orthogonality constraints of the original HFD. $\square$

*Proof of Lemma 5.* We consider that Assumption 1 is satisfied, and we apply Lemma 4 to get that

$$\eta_\ell^{(J)} = T_\ell^{(J)} \frac{f^{(J)}}{f_\ell^{(J)}} - \delta_\ell^{(J)},$$

where $\{T_\ell^{(J)}\}_J$ is the original HFD of $T_\ell(\mathbf{X})$, $\{\eta_\ell^{(J)}\}_J$ is the tree HFD of $T_\ell(\mathbf{X})$, and $\{\delta_\ell^{(J)}\}_J$ is the HFD of $\delta_\ell(\mathbf{Z}_\ell)$, with

$$\delta_\ell = \sum_{J \in \mathcal{P}_p} T_\ell^{(J)} \Big(\frac{f^{(J)}}{f_\ell^{(J)}} - 1\Big).$$

Hence, we have

$$\mathbb{E}[(\eta_\ell^{(J)}(\mathbf{X}^{(J)}) - T_\ell^{(J)}(\mathbf{X}^{(J)}))^2] = \mathbb{E}\Big[\Big(T_\ell^{(J)}(\mathbf{X}^{(J)})\Big(\frac{f^{(J)}(\mathbf{X}^{(J)})}{f_\ell^{(J)}(\mathbf{X}^{(J)})} - 1\Big) - \delta_\ell^{(J)}(\mathbf{X}^{(J)})\Big)^2\Big]$$

$$\leq \mathbb{E}\Big[T_\ell^{(J)}(\mathbf{X}^{(J)})^2 \Big(\frac{f^{(J)}(\mathbf{X}^{(J)})}{f_\ell^{(J)}(\mathbf{X}^{(J)})} - 1\Big)^2\Big] + \mathbb{E}\Big[\Big(\delta_\ell^{(J)}(\mathbf{X}^{(J)})\Big)^2\Big]$$

$$+ 2\mathbb{E}\Big[\Big|T_\ell^{(J)}(\mathbf{X}^{(J)})\Big(\frac{f^{(J)}(\mathbf{X}^{(J)})}{f_\ell^{(J)}(\mathbf{X}^{(J)})} - 1\Big)\delta_\ell^{(J)}(\mathbf{X}^{(J)})\Big|\Big].$$

We apply Cauchy-Schwartz inequality, and obtain

$$\mathbb{E}[(\eta_\ell^{(J)}(\mathbf{X}^{(J)}) - T_\ell^{(J)}(\mathbf{X}^{(J)}))^2]$$

$$\leq \left(\sqrt{\mathbb{E}\Big[T_\ell^{(J)}(\mathbf{X}^{(J)})^2 \Big(\frac{f^{(J)}(\mathbf{X}^{(J)})}{f_\ell^{(J)}(\mathbf{X}^{(J)})} - 1\Big)^2\Big]} + \sqrt{\mathbb{E}\Big[\Big(\delta_\ell^{(J)}(\mathbf{X}^{(J)})\Big)^2\Big]}\right)^2.$$

First, we bound $\mathbb{E}\Big[T_\ell^{(J)}(\mathbf{X}^{(J)})^2 \Big(\frac{f^{(J)}(\mathbf{X}^{(J)})}{f_\ell^{(J)}(\mathbf{X}^{(J)})} - 1\Big)^2\Big]$. We define $g_\ell^{(J)} = f^{(J)} - f_\ell^{(J)}$, and then,

$$\Big|\frac{f^{(J)}(\mathbf{x}^{(J)})}{f_\ell^{(J)}(\mathbf{x}^{(J)})} - 1\Big| \leq \Big|\frac{g_\ell^{(J)}(\mathbf{x}^{(J)})}{f_\ell^{(J)}(\mathbf{x}^{(J)})}\Big|.$$

Using Assumption 1, which also implies that $c_1 \leq f_\ell^{(J)}(\mathbf{x}^{(J)}) \leq c_2$, and $|g_\ell^{(J)}(\mathbf{x}^{(J)})| < \Delta_{\mathcal{A},f}$ by construction, we have

$$\frac{g_\ell^{(J)}(\mathbf{x}^{(J)})}{f_\ell^{(J)}(\mathbf{x}^{(J)})} \leq \frac{\Delta_{\mathcal{A},f}}{c_1},$$

and consequently,

$$\mathbb{E}\Big[T_\ell^{(J)}(\mathbf{X}^{(J)})^2 \Big(\frac{f^{(J)}(\mathbf{X}^{(J)})}{f_\ell^{(J)}(\mathbf{X}^{(J)})} - 1\Big)^2\Big] \leq \frac{\Delta_{\mathcal{A},f}^2}{c_1^2} \mathbb{E}[(T_\ell^{(J)}(\mathbf{X}^{(J)}))^2].$$

Next, we use Lemma 3, and get

$$\sum_{J \in \mathcal{P}_p} \mathbb{E}\Big[T_\ell^{(J)}(\mathbf{X}^{(J)})^2 \Big(\frac{f^{(J)}(\mathbf{X}^{(J)})}{f_\ell^{(J)}(\mathbf{X}^{(J)})} - 1\Big)^2\Big] \leq \Delta_{\mathcal{A},f}^2 \frac{K_1}{c_1^2} \mathbb{E}[T_\ell(\mathbf{X})^2],$$

with $K_1 = \big(1 - \sqrt{1 - \frac{c_1}{c_2}}\big)^{-2^p+1}$. Similarly, we use Lemma 3 again to get

$$\sum_{J \in \mathcal{P}_p} \mathbb{E}[\delta_\ell^{(J)}(\mathbf{Z}_\ell^{(J)})^2] \leq \Delta_{\mathcal{A},f}^2 \frac{K_1}{c_1^2} \mathbb{E}[(\sum_{J \in \mathcal{P}_p} |T_\ell^{(J)}|)^2].$$

Since

$$\mathbb{E}[\delta_\ell^{(J)}(\mathbf{Z}_\ell^{(J)})^2] = \mathbb{E}\Big[\delta_\ell^{(J)}(\mathbf{X}^{(J)})^2 \frac{f_\ell^{(J)}(\mathbf{X}^{(J)})}{f^{(J)}(\mathbf{X}^{(J)})}\Big] \geq \frac{c_1}{c_2} \mathbb{E}[\delta_\ell^{(J)}(\mathbf{X}^{(J)})^2],$$

we get

$$\sum_{J \in \mathcal{P}_p} \mathbb{E}[\delta_\ell^{(J)}(\mathbf{X}^{(J)})^2] \leq \Delta_{\mathcal{A},f}^2 \frac{c_2}{c_1^3} K_1 \mathbb{E}[(\sum_{J \in \mathcal{P}_p} |T_\ell^{(J)}(\mathbf{X}^{(J)})|)^2].$$

Then, using Cauchy-Schwartz inequality, notice that

$$\mathbb{E}\big[\big(\sum_{J \in \mathcal{P}_p} |T_\ell^{(J)}(\mathbf{X}^{(J)})|\big)^2\big] = \sum_{I,J \in \mathcal{P}_p} \mathbb{E}\big[|T_\ell^{(J)}(\mathbf{X}^{(J)})||T_\ell^{(I)}(\mathbf{X}^{(I)})|\big]$$

$$\leq \sum_{I,J \in \mathcal{P}_p} \sqrt{\mathbb{E}\big[T_\ell^{(J)}(\mathbf{X}^{(J)})^2\big]\mathbb{E}\big[T_\ell^{(I)}(\mathbf{X}^{(I)})^2\big]}$$

$$\leq 2^{2p} K_1 \mathbb{E}[T_\ell(\mathbf{X})^2].$$

We finally obtain

$$\sum_{J \in \mathcal{P}_p} \mathbb{E}[\delta_\ell^{(J)}(\mathbf{X}^{(J)})^2] \leq \Delta_{\mathcal{A},f}^2 \frac{c_2}{c_1^3} 2^{2p} K_1 \mathbb{E}[T_\ell(\mathbf{X})^2].$$

Overall, we combine the previous inequalities to state that

$$\mathbb{E}[(\eta_\ell^{(J)}(\mathbf{X}^{(J)}) - T_\ell^{(J)}(\mathbf{X}^{(J)}))^2] \leq \left(\sqrt{\Delta_{\mathcal{A},f}^2 \frac{K_1}{c_1^2} \mathbb{E}[T_\ell(\mathbf{X})^2]} + \sqrt{\Delta_{\mathcal{A},f}^2 \frac{c_2}{c_1^3} 2^{2p} K_1 \mathbb{E}[T_\ell(\mathbf{X})^2]}\right)^2$$

$$\leq \Delta_{\mathcal{A},f}^2 \frac{K_1}{c_1^2} \left(1 + 2^p \sqrt{\frac{c_2}{c_1}}\right)^2 \mathbb{E}[T_\ell(\mathbf{X})^2],$$

which gives the final result. $\qquad \square$

## C.6 Proof of Corollary 1

*Proof of Corollary 1.* We first apply Theorem 6 to get

$$\mathbb{E}[(\eta^{(J)}(\mathbf{X}^{(J)}) - m^{(J)}(\mathbf{X}^{(J)}))^2] \leq K_1 \mathbb{E}[(m(\mathbf{X}) - T(\mathbf{X}))^2] + K_2 \mathbb{E}\big[\Delta_{\mathcal{A},f}^2 \big(\sum_{\ell=1}^M \sqrt{\mathbb{E}[T_\ell(\mathbf{X})^2]}\big)^2\big],$$

where expectations are also taken with respect to the random training data of $T$ of size $n_T$. Now, we show that both terms of the above inequality converges towards zero when $n_T$ grows. For the first term, we use that $T$ is $L_2$-consistent by assumption, that is $\mathbb{E}[(m(\mathbf{X}) - T(\mathbf{X}))^2] \underset{n_T \to \infty}{\longrightarrow} 0$. Next, we analyze the second term. By definition, we have

$$\Delta_{\mathcal{A},f} = \sup_{A \in \cup_\ell \mathcal{A}_\ell} \sup_{\mathbf{x},\mathbf{x}' \in A} |f(\mathbf{x}) - f(\mathbf{x}')|.$$

Since the distribution $f$ is assumed to be a Lipchitz function with a constant that we write $K_3 > 0$, we have

$$\Delta_{\mathcal{A},f} \leq K_3 \sup_{A \in \cup_\ell \mathcal{A}_\ell} \sup_{\mathbf{x},\mathbf{x}' \in A} \|\mathbf{x} - \mathbf{x}'\|.$$

Following the notation of Scornet et al. (2015), we define the diameter of a cell $A \subset [0,1]$ by

$$\text{diam}(A) = \sup_{\mathbf{x},\mathbf{x}' \in A} \|\mathbf{x} - \mathbf{x}'\|,$$

and obtain

$$\Delta_{\mathcal{A},f} \leq K_3 \sup_{A \in \cup_\ell \mathcal{A}_\ell} \text{diam}(A).$$

By assumption, the input distribution $f$ is strictly positive on $[0,1]^p$, each node split leaves at least a fraction $\gamma > 0$ of the observations in each child node, and the optimization of node splits is slightly randomized to have a positive probability to split with all variables. Then, following (Meinshausen and Ridgeway, 2006, Lemma 2) and (Bénard et al., 2022, Lemma 5), the number of splits along each

direction grows to infinity when $n_T$ grows, and the gap between two consecutive splits cannot vanish too fast. Therefore, we get that in probability,

$$\sup_{A \in \cup_\ell \mathcal{A}_\ell} \operatorname{diam}(A) \xrightarrow[n_T \to \infty]{} 0.$$

Finally, since all trees are bounded, we obtain that the second term of the initial inequality converges towards zero, and ultimately

$$\mathbb{E}[(\eta^{(J)}(\mathbf{X}^{(J)}) - m^{(J)}(\mathbf{X}^{(J)}))^2] \xrightarrow[n_T \to \infty]{} 0.$$

$\square$

### C.7 Proof of Theorem 7

**Lemma 6.** *If Assumption 1 is satisfied, for $\beta_1^{(J)}, \ldots, \beta_{K_J}^{(J)} \in \mathbb{R}$, we have*

$$L_n \xrightarrow{p} L^\star.$$

*Proof of Theorem 7.* We consider a tree $\ell \in \mathcal{T}_M$, and that Assumption 1 is satisfied. From Lemma 6, for $\beta_1^{(J)}, \ldots, \beta_{K_J}^{(J)} \in \mathbb{R}$, we have

$$L_n \xrightarrow{p} L^\star.$$

Additionally, the set of functions $\{\eta_\ell^{(J)}\}_{J \in \mathcal{P}_p}$ is the tree HFD of $T_\ell(\mathbf{X})$, which belongs to the class of functions parameterized by $\{\beta_1^{(J)}, \ldots, \beta_{K_J}^{(J)}\}_J$, according to Theorem 2. Therefore, $L^\star$ is a convex positive function of the parameters $\{\beta_1^{(J)}, \ldots, \beta_{K_J}^{(J)}\}_J$, that has a unique minimum at $\{\eta_\ell^{(J)}\}_{J \in \mathcal{P}_p}$, where $L^\star = 0$. Since the optimization of $L_n$ is done over a compact set, the pointwise convergence above implies the uniform convergence of $L_n$ over this compact set. Finally, we can apply Theorem 5.7 from Van der Vaart (2000) to conclude that for all $J \in \mathcal{P}_p$, we have

$$\mu_{n,\ell}^{(J)} \xrightarrow{p} \eta_\ell^{(J)}.$$

$\square$

*Proof of Lemma 6.* Using the law of large numbers, we have

$$\frac{1}{n} \sum_{i=1}^n \big( T_\ell(\mathbf{X}_i) - \sum_{J \in \mathcal{P}_p} \mu_\ell^{(J)}(\mathbf{X}_i^{(J)}) \big)^2 \xrightarrow{p} \mathbb{E}\big[ \big( T_\ell(\mathbf{X}) - \sum_{J \in \mathcal{P}_p} \mu_\ell^{(J)}(\mathbf{X}^{(J)}) \big)^2 \big],$$

$$\frac{1}{n} \sum_{i=1}^n \mathbb{1}_{\mathbf{X}_i^{(J)} \in A_{k'}^{(J)} \cap \mathbf{X}_i^{(J \setminus j)} \in A_k^{(J \setminus j)}} \xrightarrow{p} \mathbb{P}(\mathbf{X}^{(J)} \in A_{k'}^{(J)} \cap \mathbf{X}^{(J \setminus j)} \in A_k^{(J \setminus j)}),$$

$$\frac{1}{n} \sum_{i=1}^n \mathbb{1}_{\mathbf{X}_i^{(J \setminus j)} \in A_k^{(J \setminus j)}} \xrightarrow{p} \mathbb{P}(\mathbf{X}^{(J \setminus j)} \in A_k^{(J \setminus j)}).$$

Then, we combine these limits to state the convergence of the empirical loss $L_n$ as follows,

$$L_n \xrightarrow{p} \mathbb{E}\big[ \big( T_\ell(\mathbf{X}) - \sum_{J \in \mathcal{P}_p} \mu_\ell^{(J)}(\mathbf{X}^{(J)}) \big)^2 \big]$$
$$+ \sum_{J \in \mathcal{P}_p} \sum_{j \in J} \sum_{k=1}^{K_{J \setminus j}} \Big[ \sqrt{\mathbb{P}(\mathbf{X}^{(J \setminus j)} \in A_k^{(J \setminus j)})} \sum_{k'=1}^{K_J} \beta_{k'}^{(J)} \mathbb{P}(\mathbf{X}^{(J)} \in A_{k'}^{(J)} | \mathbf{X}^{(J \setminus j)} \in A_k^{(J \setminus j)}) \Big]^2$$
$$= \mathbb{E}\big[ \big( T_\ell(\mathbf{X}) - \sum_{J \in \mathcal{P}_p} \mu_\ell^{(J)}(\mathbf{X}^{(J)}) \big)^2 \big]$$
$$+ \sum_{J \in \mathcal{P}_p} \sum_{j \in J} \sum_{k=1}^{K_{J \setminus j}} \mathbb{E}[\mu_\ell^{(J)}(\mathbf{X}^{(J)}) | \mathbf{X}^{(J \setminus j)} \in A_k^{(J \setminus j)}]^2 \mathbb{P}(\mathbf{X}^{(J \setminus j)} \in A_k^{(J \setminus j)}),$$

and finally, we obtain that

$$L_n \xrightarrow{p} \mathbb{E}\big[ \big( T_\ell(\mathbf{X}) - \sum_{J \in \mathcal{P}_p} \mu_\ell^{(J)}(\mathbf{X}^{(J)}) \big)^2 \big] + \sum_{J \in \mathcal{P}_p} \sum_{j \in J} \sum_{A \in \mathcal{A}^{(J \setminus j)}} \mathbb{E}[\mu_\ell^{(J)}(\mathbf{X}^{(J)}) \mathbb{1}_{\{\mathbf{X}^{(J \setminus j)} \in A\}}]^2$$
$$= L^\star.$$

$\square$

