# OpenReview forum: "Tree Ensemble Explainability through the Hoeffding Functional Decomposition and TreeHFD Algorithm"
_NeurIPS.cc/2025/Conference — NeurIPS 2025 poster_

### Official Review · Reviewer_QYXH · 2025-06-14

**Clarity:** 4
**Significance:** 3
**Originality:** 4
**Rating:** 5
**Confidence:** 4

**Summary:**

The paper develops a method to approximate the Hoeffding Functional Decomposition (HFD) of tree ensembles. The authors first derive the HFD for tree ensembles with the property that the resulting functionals are piecewise constant. They then show interesting theoretical properties of this HFD including a near hierarchical orthogonality and provide an efficient way of estimating this HFD for tree ensembles.

**Questions:**

As mentioned above, I wonder whether consistency in Theorem 7 is not very oversimplified. The theorem and proof make sense to me for a fixed partition, but as far as I understand these partitions depend on the trees and these in turn change with the sample size. So basically the HFD for tree ensembles changes with n and it is not even clear to me what the limit should be. So it seems to me that the problem is more complex than what is formalized in Theorem 7. From a statisticians perspective, I would either need more explanation here if I am wrong, or a discussion on what kind of simplyfying assumptions were taken. I think with trees it is absolutely ok to first proof constistency results in a simplified setting.

If this can be addressed sufficiently, I am happy to increase my grade.

Other than that I only have a few comments:
- I do not understand the first equality in Lines 780 and 781 (Proof of Theorem 3). Is there a short sentence that can be added to explain this?
- At the end of that proof I am missing the step where we go from \eta_{\ell} to \eta, in fact the last equations after line 786 seem to suddenly jump from \eta to \eta_{\ell}. It seems like a small step, but I think it should be explained.
- It could be good to have some description of how we get from L^* to L_n on page 6. The first part is easy to see, but the expression in L_n after the "+" appears mysterious at this point.
- Maybe a naive question from someone who does not understand computational complexity well: Why does the computational complexity not depend on n and p?
- Line 353 "hierarchical orthogonality" --> "hierarchical orthogonal"
- Detail "under the same assumptions *than* Theorem 3" sounds a bit weird do me. Maybe "as in Theorem 3"?

**Ethical Concerns:**

["NO or VERY MINOR ethics concerns only"]

**Final Justification:**

I think this is a great paper with an interesting and useful new method, and solid theory. The authors addressed my questions adequately and also replied to my proof questions in details, which was interesting to read. I also see no unresolved issue and I am thus deciding to leave my score.

**Limitations:**

Yes

**Quality:**

3

**Strengths And Weaknesses:**

Strengths
-------------------
- Very interesting and important idea.
- I enjoyed reading the paper, concepts are well explained and the proofs are overall easy to follow.
- The idea of approximate hirarchical orthogonality is very nice, as is the fact that the Hoeffding decomposition can be estimated relatively efficiently
- Great Illustrations and Experiments

Weaknesses
------------------
- My biggest concern are about details of the formalization: HFD based on Cartesian Tree Partitions are a smart move, but once we think about consistency of the approach, that somehow means to me that there is a limiting partitioning towards we converge, i.e. something like a limiting tree ensemble. I provide more details below, but if I am not wrong about this, I think this needs to be discussed.

---

> ### Author Rebuttal · Authors · 2025-07-28
>
> Thank you for your positive comments, and for taking the time to read the proofs. We are glad that you enjoyed reading the article. We will improve the clarity of the settings of the theoretical analysis thanks to your suggestions.
>
>
> **Theorem $7$.** Your comment about Theorem $7$ is highly relevant, and we realize that the settings of the theoretical analysis are not clear enough. In lines 127-129 of the article, we state that the tree ensemble is trained with a realization of the training data of fixed sample size, and that the ensemble predictor $T$ is thus considered as a deterministic function throughout the article. In fact, we mean that $T$ is trained with a set of deterministic points of fixed size (and implicitly with a fixed random seed of the tree randomizations), to obtain a deterministic function T. Then, TreeHFD is fit for $T$ using another data sample, denoted by $\mathscr{D}_n$, which is a random object, as usually formalized in learning theory.  In other words, all the theoretical results of the article are stated conditional on the training data of $T$ of fixed sample size. In Theorem $7$, we only analyze the limiting decomposition with respect to the size of $\mathscr{D}_n$, used to fit TreeHFD for a fixed function $T$ (trained with another dataset of fixed size). We do not introduce a notation for the training data of $T$ to simplify the mathematical formulations. However, we will better explain these settings to improve clarity of the theoretical analysis. Thank you for raising this important point. Also notice that in practice, we use the same dataset to train $T$ and to fit TreeHFD in the experiments, as it gives excellent performance, and splitting the data in two parts hurts accuracy. Using two datasets is a mild assumption to alleviate the theoretical analysis.
>
> To deepen the discussion for Theorem $7$, this result shows that when the size of $\mathscr{D}_n$ increases, the decomposition generated by TreeHFD converges towards the decomposition of Theorem $2$ for a given tree ensemble $T$, conditional on the training data of $T$ and the randomizations used to fit $T$. As stated in lines 142-145, the main goal of the article is to explain a given tree ensemble. Theorem $7$ is thus strongly valuable to show that the decomposition obtained by TreeHFD is well-defined by Theorem $2$, as the HFD of the function $T$ with a discretization of the orthogonality constraints. A secondary objective of the article is to show that this decomposition is a good approximation of the original HFD of the underlying regression function $m$. Hence, Theorem $6$ gives strong guarantees about the quality of this approximation. Indeed, we show that the $L_2$ error of each component is bounded by the MSE of the original tree ensemble, and the variations of the input distribution $f$ within each cell of the Cartesian tree partitions. This result is also valid conditional on the training data of $T$, and does not depend on $\mathscr{D}_n$, which is not used at this stage. Then, we can also study the accuracy of the decomposition of Theorem $2$, when the size of the data sample used to train $T$ increases. It is clear that this error converges towards zero, when the tree ensemble is consistent and the variations of $f$ within each cell  goes to zero. This last condition is verified when the diameter of each cell vanishes asymptotically [2], and the distribution $f$ is a Lipschitz function. Furthermore, the cell diameters vanish when tree depth increases with the sample size and the tree splits cannot be performed too close to the edges of cells, as explained in [1, 3].
>
> Finally, we will augment the analysis of Theorem $6$ in the article with the above discussion to strengthen the theoretical guarantees about the accuracy of TreeHFD. However, we believe that a thorough analysis of the convergence of TreeHFD towards the HFD of the regression function is out of scope of this 9-page article. Indeed, deriving sufficient conditions for the convergence of tree ensembles is a notoriously difficult problem, which is here combined with the convergence of TreeHFD. In particular, the extension of Theorem $7$ when the size of the sample used to train $T$ also increases, is highly non-trivial, and will involve a sharp analysis of the asymptotic regimes of the number of cells in the initial tree partitions and the sizes of the two data samples. We will also have to separately discuss the cases of random forests and boosted ensembles to derive realistic conditions about their training regimes. Therefore, we think that a complete analysis of this convergence deserves a separate paper.
>
>
> **Proof details.** If you consider the right hand side of the first equation line 780, $\mathbf{x}_0$ is a fixed value, and the integral is computed with respect to $\mathbf{x}^{(J)}$. Since $f$ only depends on $\mathbf{x}^{(J \setminus I)}$, we can separate the variables in the integral, and isolate the integral of $d \mathbf{x}^{(I)}$ over $A^{(I)}$, which gives the volume of this cell, and we obtain the left hand side. The same trick is used for line 781. We will add a sentence in the proof to explain this better.
>
> There is a typo in the last line of the proof of Theorem $3$. Thank you for pointing this out. For $\eta^{(I)}$, we can directly upper bound the term using $||\eta^{(I)}||$, and the $\ell$ index is a typo. However, $\eta^{(J)}$ must be broken down as the sum of $\eta_{\ell}^{(J)}$ to use the inequality of line 785, combined with the triangle inequality. Then, the term $\sum_{\ell} || \eta_{\ell}^{(J)} ||$ appears. This is a small typo in statement (ii) of Theorem $3$, which should be $\frac{c_2 - c_1}{c_1^2} ||\eta^{(I)}|| \big( \sum_{\ell} || \eta_{\ell}^{(J)} || \big) \Delta_{\mathcal{A}, f}$.
> This leaves the scope and interpretation of the result untouched, since statement (ii) shows that the gap to hierarchical orthogonality decreases with the variations $\Delta_{\mathcal{A}, f}$ of the input distribution within each cell of the Cartesian tree partitions. We will correct this in the revised version of the article.
>
>
> **Computational complexity.** The analysis of TreeHFD complexity is provided in lines 282-287 of the article. In the revised version, we will move this discussion in a separate paragraph with additional details to improve clarity, as also asked by Reviewer Ngv6. More precisely, the first step of TreeHFD is to build the constraint matrix $\mathbf{C}_n$ from the tree partitions and data sample for each tree. This is done with a linear complexity with respect to the sample size $n$, since we only need to count the number of points falling in each cell of the Cartesian tree partitions. Once this matrix is computed, the computational complexity of the optimization problem only depends on the size of the matrix $\mathbf{C}_n$, which only depends on the size of the Cartesian tree partitions, and not on the sample $n$ or dimension $p$. For example, if $p$ is very large, the number of variables and interactions involved in the decomposition of a given tree is bounded by the number of nodes in the tree, which only depends on the tree depth. Since the computational complexity is asymptotic, it does not depend on $p$.
>
>
> **Loss $L_n$.** We will further explain the connection between $L_n$ and $L^{\star}$, especially by referring to the proof of Lemma 6, where all terms are broken down.
>
>
> **Typos.** Thank you for the two typos. We will correct them in the revised article.
>
>
> **References**
>
> [1] Meinshausen, N. (2006). Quantile regression forests. Journal of Machine Learning Research, 7:983–999.
>
> [2] Scornet, E., Biau, G., and Vert, J.-P. (2015). Consistency of random forests. The Annals of Statistics, 43:1716–1741
>
> [3] Wager, S. and Athey, S. (2018). Estimation and inference of heterogeneous treatment effects using random forests. Journal of the American Statistical Association, 113:1228–1242.

---

> > ### Comment · Reviewer_QYXH · 2025-08-05
> >
> > I want to thank the authors for their detailed explanations and comments! I am happy to leave my original score unchanged.

---

### Official Review · Reviewer_Ngv6 · 2025-06-30

**Clarity:** 3
**Significance:** 3
**Originality:** 3
**Rating:** 5
**Confidence:** 3

**Summary:**

The authors introduce TreeHFD, an algorithm estimating the Hoeffding Functional Decomposition (HFD) of tree ensembles in cases where only data samples are available. HFD decomposes a function prediction into a sum of lower-order functions over variable subsets and thereby naturally enables explainability through the main effects and low-order interactions. This approach is applied by the authors to tree ensembles with unknown input distribution by discretising the orthogonality constraints. Besides theoretical analysis of the method and resulting algorithm, the authors conduct simulations and experiments with TreeHFD and establish a connection to the functional decomposition provided by TreeSHAP.

**Questions:**

- The whole model development and experimental setting revolves around regression. However, tree ensembles provide also SOTA prediction methods in the classification setting. To what extent can treeHFD be also applied/extended to this use case?
- Could treeHFD also be used for feature selection? Could it, for example, be employed to prune noisy trees with many irrelevant variables?
- Figure 2: Why are no orange points from the respective estimation with TreeSHAP included?
- Table 2: Why are there NA values for the orthogonality of the datasets "Power Plant" and "Superconduct."?
- Housing Dataset: The authors write that TreeShap does not detect the peak, however, I would argue that it does (just to lower extent than treeHFD). Can the authors elaborate on this observation? Also, how would the main effects for TreeSHAP without interactions look like in this example?

**Ethical Concerns:**

["NO or VERY MINOR ethics concerns only"]

**Final Justification:**

I keep my score

**Limitations:**

Partially. While the authors claim in the checklist that they are discussed in Section 4 with the experiments, I cannot find them there. Instead I would have preferred a separate statement on the current limitations of TreeHFD and in which scenarios it fails.

**Quality:**

3

**Strengths And Weaknesses:**

Strengths
- Interesting novel application of the HFD to tree ensembles in the practically relevant case of an available data sample but unknown input data distribution
- Extensive theoretical analysis of the HFD for tree ensembles incl. the convergence of the proposed algorithm
- Extensive experiments on both simulated and real-world data, including comparisons to relevant competitive approaches

Weaknesses
- Intuitive example for the application of treeHFD is missing. Given that it presents an XAI method to make effects of a black-box model more transparent, an example with real world data with large p and resulting interpretations would showcase its practical relevance.
- No description of the difference between interventional and observational SHAP is provided. Although is only a compared method, this would be valuable to understand the experimental section better.
- SHAP can be used to provide local explanations for a given data sample. TreeHFD could also evaluate the global functional decomposition at given inputs, however, exemplary evaluations of this scenario are not presented.
- No details on the computational complexity of the algorithm are discussed (running times, scaling to higher-order interactions etc.). Similarly, it is not discussed in which scenarios treeHFD fails.

---

> ### Author Rebuttal · Authors · 2025-07-28
>
> Thank you for your positive comments and relevant suggestions, often shared by other reviewers. We will improve the article accordingly.
>
>
> **Intuitive real examples to illustrate TreeHFD.** In section 4, we illustrate TreeHFD with a real data case, the California Housing dataset, which exhibits patterns that are easy to interpret. However, we agree that other intuitive examples should be analyzed to show the practical performance of TreeHFD, especially cases with higher dimensions and strong interactions, as also mentioned by Reviewer cnRC. In fact, we originally used three datasets to illustrate TreeHFD: the California Housing dataset, the Superconductivity dataset, and the Bike Sharing dataset. Because of the 9-page limit, we moved the results for the last two datasets to Appendix B.3.2 (pages 18-20). If the paper is accepted, we will use the additional page to move back these relevant examples to the core of the article. More precisely, the Superconductivity dataset highlights the good performance of TreeHFD for a quite large input dimension of 81, and the Bike Sharing case shows that the patterns identified by TreeHFD can be easily interpreted, as explained in the last paragraph of page 18. Overall, we believe that these examples together are convincing to show the practical efficiency of TreeHFD.
>
>
> **SHAP definitions.** The definitions of observational and interventional SHAP are briefly given in the paragraph “causal variable selection” of Section 2.2. However, there are no details in the experimental section. We will refer to the definitions of Section 2.2 in Section 4, and add complete formulas of SHAP values in Appendix B.1 to improve the understanding of the experimental section.
>
>
> **Computational complexity.** The analysis of TreeHFD complexity is provided in lines 282-287 of the article. In the revised version, we will move this discussion in a separate paragraph with additional details to improve clarity, as also suggested by Reviewer QYXH. More precisely, the first step of TreeHFD is to build the constraint matrix $\mathbf{C}_n$ from the tree partitions and data sample for each tree. This is done with a linear complexity with respect to the sample size $n$, since we only need to count the number of points falling in each cell of the Cartesian tree partitions. Once this matrix is computed, the computational complexity of the optimization problem only depends on the size of the matrix $\mathbf{C}_n$, which only depends on the size of the Cartesian tree partitions, and not on the sample $n$ or dimension $p$. For example, if $p$ is very large, the number of variables and interactions involved in the decomposition of a given tree is bounded by the number of nodes in the tree, which only depends on the tree depth. We highlight that $\mathbf{C}_n$ is highly sparse by design, and we can therefore use efficient solvers of sparse QP problems, such as $\texttt{osqp}$ package. Finally, notice that we provide an example of running time of TreeHFD in Appendix B.1 (lines 533-537).
>
>
> **Separate limitation paragraph.** The limitations of the introduced TreeHFD algorithm are spread across all sections. We agree that it is better to write a specific paragraph to group the main limitations, especially about the inefficiency of TreeHFD for large tree depths (also mentioned by Reviewer cnRC), and the required access to the internal tree structures (mentioned by Reviewer iKso). Thank you for the suggestion.
>
>
> **Local explanations.** TreeHFD can explain a given prediction similarly to SHAP, by the break down of the prediction across all input variables and interactions, which can be displayed efficiently through a waterfall plot. We agree that it would be interesting to present such cases in the article. However, such results over few prediction points only illustrate the algorithms, but cannot be used to assess performance. Because of the page limit, we would rather prioritize the global evaluations of Tables 1 and 2, and add additional real datasets in the potential additional page.
>
>
> **Classification case.** The extension of TreeHFD for XGBoost models to classification problems is straightforward. Indeed, the predicted logits for each class are continuous outputs, and can therefore be handled as the regression case to obtain the HFD of the logit for each class. However, in the case of random forests, classification problems are solved using majority votes, where each tree outputs a predicted class. In this case, it seems that the HFD defined by Stone and Hooker does not directly apply. The specific case of binary classification can be treated as a regression problem, and TreeHFD naturally applies in this case for all tree ensembles. Hence, we will also add this discussion in the revised version of the article.
>
>
> **Feature selection.** TreeHFD can be used for feature selection. For example, an importance measure can be defined for each variable and interaction through the variance of the associated decomposition component. Then, this feature importance can be used for variable selection, when combined with uncertainty quantification methods, to discard irrelevant features. However, such extension is out of scope of the article given the large literature about variable importance and uncertainty quantification, but is definitely an interesting idea for future work.
>
>
> **Details for figures and tables.**  The NA in Table 2 are explained in the third paragraph of Appendix B.3.1 (lines 645-654). In fact, we only consider interactions with a non-negligible variance, otherwise the correlation coefficient is undefined. For these datasets with NA, there is no interactions with the chosen threshold of 1% of the output variance.
>
> In Figure 2, TreeSHAP is displayed in red. The solid lines in red and orange are the theoretical decomposition components derived from the value functions of respectively interventional and observational SHAP. There is only one version of TreeSHAP algorithm, designed from a heuristic procedure, which does not clearly target interventional or observational SHAP. In our experiments, we show that the decomposition induced by TreeSHAP is closer to the HFD than the two SHAP decompositions. We are not sure that we have properly understood this comment. Let us know if it is not the case.
>
> For the peak of TreeSHAP for the Housing dataset, we agree that TreeSHAP slightly detects it. However, the peak is small compared to local noise of TreeSHAP decomposition, and it is therefore hard to understand whether it is significant. On the other hand, it is clear for TreeHFD. We will soften this comment in the article, following your remark.

---

> > ### Comment · Reviewer_Ngv6 · 2025-08-05
> >
> > I would like to thank the authors for their responses to my comments

---

### Official Review · Reviewer_iKso · 2025-07-03

**Clarity:** 3
**Significance:** 2
**Originality:** 2
**Rating:** 4
**Confidence:** 3

**Summary:**

The paper introduces TreeHFD, a post-hoc explanation method for tree ensemble models that estimates the Hoeffding Functional Decomposition (HFD) directly from data without requiring knowledge of the input distribution. HFD breaks the model prediction into a sum of orthogonal components over variable subsets, enabling interpretable representations of main effects and interactions. TreeHFD ensures key properties like uniqueness, sparsity, near-orthogonality, and causal variable selection. It outperforms TreeSHAP in terms of stability and interpretability, and the authors show empirically that TreeSHAP closely approximates the HFD but suffers from entangled and noisy components.

**Questions:**

The paper’s focus is restricted to regression problems, where the output variable is continuous and modeled as $Y = m(X) + \epsilon$. How could it be applied to the classification tasks?

**Ethical Concerns:**

["NO or VERY MINOR ethics concerns only"]

**Final Justification:**

I think the paper puts forward an interesting way of computing Hoefdding decomposition (also known as functional ANOVA decomposition) for tree ensemble models, which is critical in interpretable machine learning.

**Limitations:**

TreeHFD operates directly on the structure of the trained tree ensemble and thus requires full access to the internal components of each tree, including split variables, split thresholds, and leaf values. It is not a huge limitation in explainability, but should be briefly discussed.

**Paper Formatting Concerns:**

Nothing specific

**Quality:**

3

**Strengths And Weaknesses:**

The paper studies an important aspect in explainability by fining the Hoeffding functional decomposition for dependent data. The Hoeffding functional decomposition is a functional decomposition with extra constraints of zero-mean functions and orthogonality; this paper puts forward a way to construct such a decomposition for tree ensembles in polynomial time, which is a drastic improvement over the recursive construction in exponential time. That is why the paper is of significant importance.

A key limitation of TreeHFD is its scalability with respect to the complexity of the trees in the ensemble.  Although the algorithm is efficient in terms of sample size, the complexity of the decomposition grows with the tree depth, the number of interaction terms, and the number of splits, which can become computationally intensive for deep or large ensembles unless pruning is applied. To keep the decomposition tractable, the authors restrict the order of interactions to at most two by setting the interaction hyperparameter d_I =2, meaning that only main effects and pairwise interactions are modeled. This setting seems to be consistently used across all experiments in the paper. While this promotes interpretability and computational efficiency, it may overlook higher-order interactions that could be relevant in some domains, and is in contrast with the efficiency claims in the paper.

In addition, the paper assumes that input features lie within the unit hypercube $[0,1]^p$ with a density bounded away from zero and infinity. While this assumption simplifies the theoretical analysis—ensuring well-behaved integrals, uniqueness of the decomposition, and control over orthogonality—it does not naturally hold in many real-world datasets where features vary in scale or distribution. Although normalization can map features to the $[0,1]$ range, this transformation may not have been used in the original trained model and   could obscure relationships in the original feature space. Moreover, the assumption of a uniformly bounded density may be unrealistic in high-dimensional or highly skewed settings, potentially limiting the robustness of the decomposition and its convergence guarantees.

Another limitation lies in the way the paper assumes the tree ensemble prediction function is constructed as a simple sum of the outputs from individual trees, i.e., $T = \sum_\ell T_\ell$. While this assumption holds in some models like random forests (up to averaging), it does not directly apply to boosting algorithms like XGBoost, where each tree output is scaled by a learning rate. This assumption appears throughout the main theorems and derivations, yet the paper does not explicitly clarify whether these scaling factors are absorbed into the definition of $T_\ell$ or how they impact the theoretical validity of the decomposition. Clarifying this point seems to be essential for a broader applicability of the paper.

For readers unfamiliar with the Hoeffding Functional Decomposition (HFD), the paper could benefit from a brief exposition of the classical case—where input features are assumed to be independent—and the recursive formulation used to construct the decomposition. This would not only make the paper more accessible to a broader audience, but also highlight the difficulty of constructing HFD even for independent features.

I am willing to raise my score if these concerns are adequately addressed.

---

> ### Author Rebuttal · Authors · 2025-07-25
>
> Thank you for the detailed review, which provides many relevant comments and suggestions. This will help us improve the article, especially the clarity of several points.
>
> **Ensemble defined as the sum of individual trees.** The scaling factors involved in the aggregation of trees (e.g., inverse of number of trees in random forests, and learning rate for boosted ensembles) are indeed absorbed in the definition of each tree predictor $T_{\ell}$. This is stated lines 130-132 of the article. Therefore, all the developed theory applies to xgboost tree ensembles. We will rephrase these sentences to improve clarity of this definition.
>
>
> **HFD for independent inputs.** We agree that a brief presentation of the HFD for independent variables will improve the accessibility of the article. We will include it in Section 2.1 of the revised  article. Thank you for this suggestion.
>
>
> **Assumptions for the input distribution.** We assume that the input variables lie in the unit hypercube, as is often done in the theoretical analysis of tree ensembles [3, 5]. Importantly, the unit cube is chosen for its simplicity and without loss of generality, since all the developed theory also hold for any hyperrectangle support. Indeed, the HFD for dependent inputs stated in Theorem 1 holds for any hyperrectangle support, as explained at the end of page 126 of [4], and in Theorem 1 from [1]. This is also the case for our theoretical analysis.
> In particular, datasets with features that greatly vary in scale are included in this assumption. In fact, this assumption of bounded support is not very restrictive. Finally, we agree that the assumption of positive bounds of the distribution is slightly more restrictive, but strongly alleviates the mathematical analysis. We will add these comments in the revised version of the article.
>
>
> **Classification case.** The extension of TreeHFD for XGBoost models to classification problems is straightforward. Indeed, the predicted logits for each class are continuous outputs, and can therefore be handled as the regression case to obtain the HFD of the logit for each class. However, in the case of random forests, classification problems are solved using majority votes, where each tree outputs a predicted class. In this case, it seems that the HFD defined by Stone and Hooker does not directly apply. The specific case of binary classification can be treated as a regression problem, and TreeHFD naturally applies in this case for all tree ensembles. Hence, we will also add this discussion in the revised version of the article. Thank you again for the suggestion.
>
>
> **Access to the internal tree structures.** Indeed, TreeHFD needs to access the internal structure of the trees, which is rarely a limitation, since all main software packages for tree ensembles provide an API to retrieve tree structures. It may not be possible in specific cases, such as pre-trained models or proprietary tools. We will add this comment in a dedicated limitation paragraph.
>
>
> **Order of interactions.** We only include main effects and second-order interactions in TreeHFD decompositions, since they are sufficient to break down the initial xgboost models for all tested datasets, as shown by the small residuals in Table 2. If relevant for specific cases, we could easily add third-order interactions in TreeHFD for shallow trees. However, for higher-order interactions, the Cartesian tree partitions may suffer from the curse of dimensionality. Overall, notice that interactions of order three or higher cannot be represented, and are therefore not really useful to build explainable models. Consequently, we focus the experimental section on main effects and second-order interactions. We will further emphasize the limitation of TreeHFD for high-order interactions in the article.
>
>
> **Computational complexity of TreeHFD for large tree depths.** The TreeHFD algorithm has a high computational complexity when the tree depth is large,  as also highlighted by Reviewer cnRC. This is clearly stated in lines 294-296 of the article, and further details are provided in Section A of the Appendix. If the article is accepted, we will take advantage of the additional page to move this important discussion to the core of the article.
>
> XGBoost tree ensembles most often use shallow trees, with a depth between 3 and 8, which is easily handled by TreeHFD. The second main type of tree ensembles is random forests, which use fully-grown trees, where TreeHFD may indeed become inefficient. However, it is frequently mentioned in the literature that splits at the bottom of trees are not significant in random forests, and that reducing tree depth barely impacts accuracy [2]. The main convergence result of random forests does not hold for fully-grown trees, but requires an asymptotic regime where the number of terminal leaves is not too large with respect to the sample size [3]. From a practical point of view, we compare the accuracy of random forests with default parameters to the case where tree depth is limited to 12, for all real datasets of Section 4. We display the performance in the table below, and observe that the loss of accuracy is very small in all cases, except two (accuracy is defined as the proportion of output explained variance). This shows that the required pruning preprocessing of TreeHFD for tree ensembles with large depths is not a strong limitation, although it may indeed induce a small reduction of forest accuracy in specific cases.
>
> | Dataset | Tree depth (Fully-grown RF) | Accuracy (Fully-grown RF) |  Accuracy (RF with tree depth of 12) |
> |:----------:|:--------------------------------------:|:-----------------------------------:|:-------------------------------------------------:|
> | Abalone | 33 | 0.56 | 0.56 |
> | Airfoil | 14 | 0.90 | 0.90 |
> | Bike Sharing | 33 | 0.81 | 0.74 |
> | Housing | 31 | 0.81 | 0.78 |
> | Concrete | 22 | 0.91 | 0.90 |
> | Nutrition | 28 | 0.32 | 0.33 |
> | Parkinson | 34 | 0.79 | 0.74 |
> | Power Plant | 18 | 0.97 | 0.96 |
> | Superconductivity | 49 | 0.93 | 0.91 |
>
>
> **References**
>
> [1] Chastaing, G., Gamboa, F., and Prieur, C. (2012). Generalized Hoeffding-Sobol decomposition for dependent variables - application to sensitivity analysis. Electronic Journal of Statistics, 6:2420 – 2448.
>
> [2] Duroux, R., & Scornet, E. (2018). Impact of subsampling and tree depth on random forests. ESAIM: Probability and Statistics, 22:96-128.
>
> [3] Scornet, E., Biau, G., and Vert, J.-P. (2015). Consistency of random forests. The Annals of Statistics, 43:1716–1741.
>
> [4] Stone, C. J. (1994). The use of polynomial splines and their tensor products in multivariate function estimation. The Annals of Statistics, 22:118–171.
>
> [5] Wager, S. and Athey, S. (2018). Estimation and inference of heterogeneous treatment effects using random forests. Journal of the American Statistical Association, 113:1228–1242.

---

> > ### Comment · Reviewer_iKso · 2025-08-04
> > **response**
> >
> > would like to thank the authors for their responses.
> >
> > However, my two primary concerns remain unaddressed, both of which pertain to the applicability of the proposed method. First, while the paper aims to explain tree ensemble models, the assumption that the input distribution lies within a hypercube is quite restrictive. In practice, models are not typically constructed under this constraint, which raises concerns about the method’s real-world applicability.
> >
> > Second, the time complexity of the approach is a critical issue. The core contribution of the paper appears to be an efficient computation of the Hoeffding decomposition, but the high computational cost undermines this goal and further limits the method’s practical utility.

---

> > > ### Author Response · Authors · 2025-08-04
> > >
> > > Thank you for reading our rebuttal, and taking the time to give your feedback. We have critical comments to add regarding the input distribution support and the computational complexity. We hope you can find the time to read them.
> > >
> > > **Input distribution support.**
> > > The assumption of hyperrectangle support for the input distribution $f$ is only required to state Theorem $6$ with the HFD of $m$, which is not the main result. The tree HFD of $T$ given in Theorem $2$ does not require this assumption. Indeed, by construction of the tree ensemble $T$, all terminal leaves of each tree contain at least one data point, which means that the input distribution averaged over each cell of a tree partition is strictly positive. Additionally, since the number of terminal leaves is finite, the tree ensemble is constant in all directions outside an hyperrectangle, and the tree HFD of $T$ is thus also constant outside this hyperrectangle. Inside the hyperrectangle, the input distribution averaged over each cell of a tree partition, automatically satisfies this assumption of hyperrectangle support with bounded values away from zero, even if it is not the case of the original input distribution $f$. Therefore, Theorem $2$ applies for input distributions with unbounded supports that take null values. To avoid a discrepancy with Theorem $6$, we initially chose to state all results in the context of Assumption $1$. We will add this discussion to show that TreeHFD is valid in general settings.
> > >
> > > **Computational complexity.**
> > > The core of the article contribution is to design an algorithm to estimate the HFD of tree ensembles with dependent inputs, since no algorithm exists to achieve this task, to our best knowledge. Additionally, TreeHFD is a fast algorithm for xgboost models and random forests with limited depths. In the specific case of fully-grown forests, TreeHFD becomes computationally costly, but it is possible to apply a pruning step to recover a fast algorithm, with a very small impact on accuracy in most cases.

---

> > > > ### Comment · Reviewer_iKso · 2025-08-05
> > > > **response2**
> > > >
> > > > I would like to thank the authors for further clarification, and I will raise my score. Nonetheless, as a reader, I would like to see some of the discussions here in the main body of the paper, given that there is an extra page in the camera-ready version.

---

> > > > > ### Author Response · Authors · 2025-08-05
> > > > >
> > > > > Thank you for your multiple answers, and for revising your review. Yes, we will add the above discussion in the core of the article for clarification.

---

### Official Review · Reviewer_cnRC · 2025-07-03

**Clarity:** 3
**Significance:** 3
**Originality:** 3
**Rating:** 4
**Confidence:** 2

**Summary:**

This paper introduces TreeHFD, an algorithm designed to compute the Hoeffding Functional Decomposition (HFD) for tree ensemble models such as gradient boosted trees and random forests. The HFD is an explainability method that provides a unique decomposition of a function into a sum of main effects (univariate functions) and interaction terms (multivariate functions). However, its practical estimation, particularly in the common machine learning setting with dependent input variables and an unknown data distribution, is a challenging task. The authors propose an algorithm that takes a trained tree ensemble and a dataset as input, and formulate a loss function for each tree composed of two parts: a mean squared error term to ensure the decomposition accurately reconstructs the tree's predictions, and a penalty term to enforce the hierarchical orthogonality that defines pure interactions. This objective is then solved as a standard least-squares problem. The authors provide theoretical justification for their approach, namely the uniqueness of their proposed TreeHFD, the near-orthogonality of the final components, the sparsity of the decomposition, the algorithm's convergence, and its ability to perform causal variable selection. They also empirically demonstrate that TreeHFD accurately estimates the HFD on synthetic data.

**Questions:**

1. Please expand the analysis on at least one or two more datasets, showing how TreeHFD captures main effects and interactions.

**Ethical Concerns:**

["NO or VERY MINOR ethics concerns only"]

**Final Justification:**

The authors answered all my questions and concerns. However, I am still concerned that a fixed tree depth of 12 might be a restrictive hyperparameter for tasks with greater complexity. Given strengths I previously mentioned, I will maintain my original score.

**Limitations:**

Mentioned in the weaknesses:
The algorithm's computational cost is dependent on the number of cells in the Cartesian tree partition, which can increase exponentially with tree depth. The authors propose using hyperparameters ($d_T$, $d_V$) to prune deep trees or limit the number of variables considered as a practical solution. This approach, however, feels somewhat ad-hoc. The paper uses XGBoost as an ensemble (where the default depth is 6) , but for other ensembles that require deep trees, such as those in a standard Random Forest, this pruning could lead to accuracy degradation.

**Quality:**

3

**Strengths And Weaknesses:**

Strengths:
- The authors define the HFD specifically for tree ensembles, introducing a "Cartesian tree partition" input space splitting, which helps with the discretization of the orthogonality constraints.
- The proposed work is grounded on mild assumptions, namely that the input distribution has bounded density on a hyperrectangle support, which is often a reasonable assumption for tree-based models.
- The proposed TreeHFD algorithm is not merely a heuristic. The authors define a TreeHFD by adapting the classic HFD framework to the piecewise-constant nature of tree-based models. They formally derive its essential properties, such as uniqueness, orthogonality, sparsity, and causal variable selection.
- The experimental evaluation is well-structured. The use of an analytical case with a known ground-truth HFD provides a clear demonstration of TreeHFD's accuracy. The qualitative analysis of the California Housing dataset coincides with common sense, although it might benefit from more analysis on the other variables and their interactions.

Weaknesses:
- The algorithm's computational cost is dependent on the number of cells in the Cartesian tree partition, which can increase exponentially with tree depth. The authors propose using hyperparameters ($d_T$, $d_V$) to prune deep trees or limit the number of variables considered as a practical solution. This approach, however, feels somewhat ad-hoc. The paper uses XGBoost as an ensemble (where the default depth is 6) , but for other ensembles that require deep trees, such as those in a standard Random Forest, this pruning could lead to accuracy degradation.

---

> ### Author Rebuttal · Authors · 2025-07-25
>
> Thank you for taking the time to write this review, and for the relevant comments and suggestions. In particular, the question about additional datasets will clearly improve the article. We also address the comment about the computational complexity of TreeHFD.
>
>
>
> **Additional real datasets to illustrate TreeHFD.** We agree that analyzing several real data cases is essential to show the practical performance of an XAI method. In fact, we originally used three datasets to illustrate TreeHFD: the California Housing dataset, the Superconductivity dataset, and the Bike Sharing dataset. Because of the 9-page limit, we moved the results for the last two datasets to Appendix B.3.2 (pages 18-20). If the paper is accepted, we will use the additional page to move these relevant examples back to the core of the article. More precisely, the Superconductivity dataset highlights the good performance of TreeHFD for a quite large input dimension of 81, and the Bike Sharing case shows that the patterns identified by TreeHFD can be easily interpreted, as explained in the last paragraph of page 18.
>
>
> **Computational complexity of TreeHFD for large tree depths.** The TreeHFD algorithm has a high computational complexity when the tree depth is large,  as also highlighted by Reviewer iKso. This is clearly stated in lines 294-296 of the article, and further details are provided in Section A of the Appendix. We will also take advantage of a potential additional page to move this important discussion to the core of the article.
>
> XGBoost tree ensembles most often use shallow trees, with a depth between 3 and 8, which is easily handled by TreeHFD. The second main type of tree ensembles is random forests, which use fully-grown trees, where TreeHFD may indeed become inefficient. However, it is frequently mentioned in the literature that splits at the bottom of trees are not significant in random forests, and that reducing tree depth barely impacts accuracy [1]. The main convergence result of random forests does not hold for fully-grown trees, but requires an asymptotic regime where the number of terminal leaves is not too large with respect to the sample size [2]. From a practical point of view, we compare the accuracy of random forests with default parameters to the case where tree depth is limited to 12, for all real datasets of Section 4. We display the performance in the table below, and observe that the loss of accuracy is very small in all cases, except two (accuracy is defined as the proportion of output explained variance). This shows that the required pruning preprocessing of TreeHFD for tree ensembles with large depths is not a strong limitation, although it may indeed induce a small reduction of forest accuracy in specific cases.
>
> | Dataset | Tree depth (Fully-grown RF) | Accuracy (Fully-grown RF) |  Accuracy (RF with tree depth of 12) |
> |:----------:|:--------------------------------------:|:-----------------------------------:|:-------------------------------------------------:|
> | Abalone | 33 | 0.56 | 0.56 |
> | Airfoil | 14 | 0.90 | 0.90 |
> | Bike Sharing | 33 | 0.81 | 0.74 |
> | Housing | 31 | 0.81 | 0.78 |
> | Concrete | 22 | 0.91 | 0.90 |
> | Nutrition | 28 | 0.32 | 0.33 |
> | Parkinson | 34 | 0.79 | 0.74 |
> | Power Plant | 18 | 0.97 | 0.96 |
> | Superconductivity | 49 | 0.93 | 0.91 |
>
>
> **References**
>
> [1] Duroux, R., & Scornet, E. (2018). Impact of subsampling and tree depth on random forests. ESAIM: Probability and Statistics, 22:96-128.
>
> [2] Scornet, E., Biau, G., and Vert, J.-P. (2015). Consistency of random forests. The Annals of Statistics, 43:1716–1741.

---

> > ### Comment · Reviewer_cnRC · 2025-08-08
> >
> > I thank the authors for their response and additional experiments. I will keep my score.

---

### Decision · Program_Chairs · 2025-09-17

**Decision:**

Accept (poster)

**Comment:**

The paper introduces a new algorithm for decomposing the fits of regression tree ensembles (outputted, e.g., by random crests or XGBoost) into a sum of main effects (i.e., univariate functions) and interaction effects (i.e., bivariate functions). The authors shows that the proposed method enjoys favorable theoretical properties and establish an interesting link to the popular TreeSHAP heuristic.

All reviewers agreed that the paper was interesting and had the potential to be impactful. They initially sought clarification on several points related to the scalability and computational complexity (iKSO & Ngv6), the need to access internal tree structures, and the relevance/importance of Theorem 7 (Reviewer QYXH). The reviewers and I found the authors' responses satisfactory but would encourage the authors to make the following revisions to the paper:

  1. Include a summary of the discussion with iKSO about the assumptions surrounding the input distribution and complexity
  2. Clarify the setting of Theorem 7.

I additionally suggest including plain-language summaries of all theoretical results. In the current version, theorems often appear without being cross-referenced or discussed in the preceding exposition. Space permitting, some sign-posting would be helpful for readers.